# How Many Images Does It Take?
# Estimating Imitation Thresholds in Text-to-Image Models

Sahil Verma[1]    Royi Rassin[2]    Arnav Das[*1]    Gantavya Bhatt[*1]    Preethi Seshadri[*3]
Chirag Shah[1]    Jeff Bilmes[1]    Hannaneh Hajishirzi[1,4]    Yanai Elazar[2]

[1]University of Washington, Seattle    [2]Bar-Ilan University    [3]University of California, Irvine    [4]Allen Institute of AI

Reviewed on OpenReview: https://openreview.net/forum?id=x0qJo7SPhs

## Abstract

Text-to-image models are trained using large datasets of image-text pairs collected from the internet. These datasets often include copyrighted and private images. Training models on such datasets enables them to generate images that might violate copyright laws and individual privacy. This phenomenon is termed *imitation* – generation of images with content that has recognizable similarity to its training images. In this work we estimate the point at which a model was trained on enough instances of a concept to be able to imitate it – the *imitation threshold*. We posit this question as a new problem and propose an efficient approach that estimates the imitation threshold without incurring the colossal cost of training these models from scratch. We experiment with two domains – human faces and art styles, and evaluate four text-to-image models that were trained on three pretraining datasets. We estimate the *imitation threshold* of these models to be in the range of 200-700 images, depending on the domain and the model. The *imitation threshold* provides an empirical basis for copyright violation claims and acts as a guiding principle for text-to-image model developers that aim to comply with copyright and privacy laws. Website: https://how-many-van-goghs-does-it-take.github.io/. Code: https://github.com/vsahil/MIMETIC-2.

## 1 Introduction

The progress of text-to-image models in recent years (Ramesh et al., 2021; Rombach et al., 2022; Goodfellow et al., 2022; Yao et al., 2025) is much attributed to the availability of large-scale pretraining datasets like LAION (Schuhmann et al., 2022). These datasets consist of semi-curated image-text pairs scraped from Common Crawl, which leads to the inclusion of explicit, copyrighted, and private material (Cavna, 2023; Hunter, 2023; Vincent, 2023; Jiang et al., 2023; Birhane et al., 2021). Training models on such images may be problematic because text-to-image models can *imitate* — generate images with highly recognizable features from their training data (Somepalli et al., 2023a; Carlini et al., 2023a). This behavior has both legal and ethical implications, such as copyright infringements as well as privacy violations of individuals whose images are present in the training data without consent, that has led to several lawsuits by artists against such model providers (Saveri & Butterick, 2023).

Previous work proposed methods for detecting when models memorize training images (Somepalli et al., 2023a; Carlini et al., 2023a), and mitigation techniques (Somepalli et al., 2023b; Shan et al., 2023). For instance, researchers found that duplicate images increase the chance of memorization. Typically, *memorization* refers to the replication of a specific training image. Instead of measuring *memorization*, we focus on *imitation* - a broader and under-explored sense of memorization, where a concept is recognizable from a generated image.

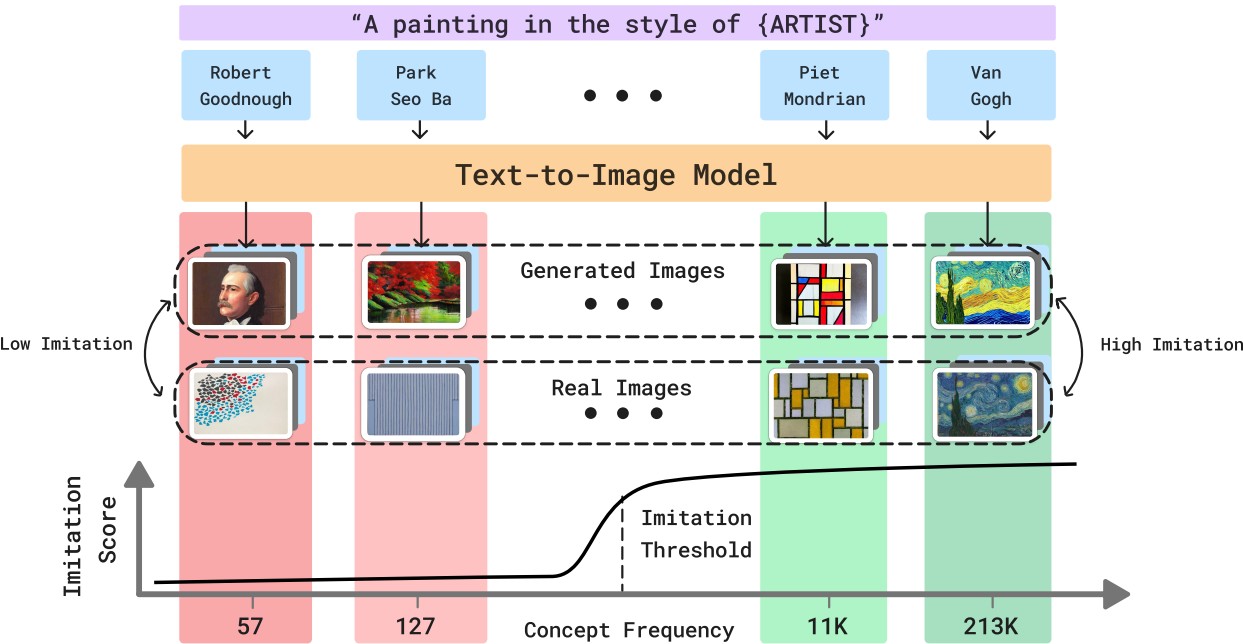

Figure 1: An overview of the imitation phenomenon where we seek the *imitation threshold* – the point at which a model was exposed to enough instances of a concept that it can reliably imitate it. The figure shows four concepts (e.g., Van Gogh's art style) that have different counts in the training data (e.g., 213K for Van Gogh). As the image count of a concept increases, the ability of the text-to-image model to imitate it increases (e.g., Piet Mondrian's and Van Gogh's art styles have higher imitation). The *imitation threshold* represent the number of instances a model has to be trained on such that humans recognize such concept in generated images.

In this work, we ask **how many instances of a concept does a text-to-image model need to be trained on to imitate it**, where concept can refer to a specific person or a specific art style, for example. Establishing such an *imitation threshold* is useful for several reasons. First, it offers an empirical basis for copyright and privacy violation claims, suggesting that if a concept's prevalence in the training data is below this threshold, a model trained on such data cannot reproduce such concept (Saveri & Butterick, 2023; Vincent, 2023; Ceoln, 2022). Second, it acts as a guiding principle for text-to-image models developers that want to avoid such violations. Finally, it reveals an interesting connection between training data statistics and model behavior, and the ability of models to efficiently harness training data (Udandarao et al., 2024; Carlini et al., 2023b). We posit this question as a new problem, and provide its schematic overview in Figure 1.

To find the gold imitation threshold one has to train counterfactual models while varying the number of images of a concept and measuring models' imitation abilities. However, training even one of these models is extremely expensive (Bastian, 2022). Instead, we propose a tractable alternative, **M**easuring **Im**itation Thr**E**shold **T**hrough **I**nstance **C**ount and **C**omparison (MIMETIC$^2$), that estimates the threshold without training multiple models using observational data. We start by collecting a large set of concepts (e.g., various kinds of art styles) per domain (e.g., domain of art styles), and use a text-to-image model to generate images for each concept. Then, we compute the frequency (in the training data) and the imitation score of each concept (§4.1). Then, we estimate the imitation scores using dedicated embeddings that capture the similarities of concepts (§4.2). Finally, we estimate the imitation threshold for each domain using a *change detection* algorithm (Killick et al., 2012) (§4.3).

Operating with observational data, a naive approach may lead to a biased estimate as confounding variables such as the quality of the imitation scoring model on different groups within the domain, or estimating the training frequencies of concepts (e.g., simple counts of 'Van Gogh' in the captions results in a biased estimate since the artist may be mentioned in the caption without their painting in the corresponding image). We carefully tailor MIMETIC$^2$ to deconfound such variables (§4). Since our approach is observational in nature, we also conduct counterfactual experiments to validate our results (§C). We train models on new concepts that were not seen in the training data, and vary the concept frequency to several values both smaller and

larger than our estimated imitation threshold. These experiments demonstrate that our estimated imitation thresholds are accurate (with a difference of upto 10% in the estimated thresholds) .

Overall, we propose and formalize a new problem that estimate the number of images of a concept that a text-to-image model requires for imitating it (§3), and propose a method MIMETIC$^2$ that efficiently estimates the *imitation threshold* for these models (§4). We apply our approach to the domains of human face and art style imitation on four text-to-image models and estimate the imitation thresholds for such models to be in the range of 200-700 images (§6). These imitation thresholds provide concrete insights on imitation abilities, can act as an empirical basis for copyright violation claims, and as guiding principles for model developers.

## 2 Background

**Text-to-Image Models** Generating images from textual inputs is a long studied problem, with diffusion models as the current state-of-the-art (Sohl-Dickstein et al., 2015; Dhariwal & Nichol, 2021).

Diffusion models are generative models that learn to approximate the underlying data distribution. Given a trained diffusion model, it is possible to sample synthetic images from the learned distribution by performing a sequence of denoising operations. Open-sourced models like Stable Diffusion (SD) (Rombach et al., 2022) are trained using large datasets such as LAION (Schuhmann et al., 2022), a large image-caption dataset scraped from Common Crawl.

**Dataset Issues and Privacy Violations** The advancement in text-to-image capabilities, largely due to large datasets, is accompanied by concerns about the training on explicit, copyrighted, and private material (Birhane et al., 2021) and imitating such content when generating images (Cavna, 2023; Hunter, 2023; Vincent, 2023; Jiang et al., 2023; He et al., 2024). For example, (Birhane et al., 2021) and (Thiel, 2023) found several explicit images in the LAION dataset, and Getty Images found millions of their copyrighted images in LAION (Vincent, 2023). Issues around imitation of training images has especially plagued artists, whose livelihood is threatened (Saveri & Butterick, 2023; Shan et al., 2023), as well as individuals whose face has been used without consent to create inappropriate content (Hunter, 2023; Badshah, 2024). The imitation threshold would be a useful basis for copyright infringement and privacy violation claims in such cases.

**Training Data Statistics and Model Behavior** Pretraining datasets are a core factor for explaining model behavior (Elazar et al., 2024). (Razeghi et al., 2022) found that the few-shot performance of language models is highly correlated with the frequency of instances in pretraining dataset. (Udandarao et al., 2024) bolster this finding by demonstrating that the performance of multimodal models on downstream tasks is strongly correlated with a concept's frequency in the pretraining datasets. In addition, (Carlini et al., 2023b) show that language models more easily memorize highly duplicated sequences. As such, expanding the duplicates finding to the same concept re-appearing in different images, we can intuitively conjecture such concepts will be learned and then imitated by models. In this work, we study this question, and quantify the number of images to appear in the training data that are required to imitate a concept.

## 3 Problem Formulation and Overview

In this work, we seek to find the minimal number of images of a *concept* a text-to-image model requires in order to imitate it.

**Defining Concept:** We follow the same definition of concepts as previous work (Udandarao et al., 2024): we consider specific instances of human faces and artist styles as distinct concepts. As such, each item in a domain constitutes a concept; human individuals and artistic styles in the *human face* and *art style* domains, respectively. While there might be some visual similarities between concepts, especially for art styles, we employ discriminator models that distinguish between concepts with high accuracy (which is necessary for the accurate estimation of the imitation threshold).

**Setup:** Our setup involves a training dataset $\mathcal{D} \triangleq \{(\mathbf{x_i}, \mathbf{y_i}) \mid \mathbf{x_i} \in \mathcal{X}, \ \mathbf{y_i} \in \mathcal{Y}\}_{i=1}^{n}$, composed of $n$ (image, caption) pairs, where $\mathcal{X}$ and $\mathcal{Y}$ represents space of images and text captions, respectively, and a model

$\mathcal{M}$ that is trained on $\mathcal{D}$ to generate an image $\mathbf{x}$ given the text-caption $\mathbf{y}$. $\mathcal{M}(\mathbf{y})$ denotes the generated image for a provided caption $\mathbf{y}$. Let $\mathbb{I}^j(\mathbf{x})$ be an indicator function that indicates whether a concept $Z^j$ is present in an image $\mathbf{x}$. Each concept $Z^j$ appears $c^j = \sum_i \mathbb{I}^j(\mathbf{x_i})$ times in the dataset $\mathcal{D}$, where the dataset may contain multiple concepts $\{Z^1, Z^2, \dots\}$. Lastly, $\mathcal{M}_k^j$ denotes a model trained on a dataset where $c^j = k$.

The *imitation threshold* is the minimal number of training images containing a concept $Z^j$ that model $\mathcal{M}$ generates images $\mathcal{M}(\mathbf{p}^j)$ (with different random seeds) from a prompt $\mathbf{p}^j$ which mentions the concept $Z^j$, and $Z^j$ is visually recognizable. [1] For example, if $Z^j$ refers to Van Gogh's art style, then the imitation threshold is the minimal number of training images of Van Gogh's paintings in a dataset used to train a text-to-image model, for which the model can generate images that imitates Van Gogh's art style. We consider a generated image to be an imitation of a concept if the similarity between the generated image and the original images of that concept in the training data is above a threshold. We measure the similarity using a concept specific detector model and we determine the detection threshold by conducting experiments on the original images of the concept (Section 4.2).

$$\text{Imitation Threshold}^j \triangleq \min \left\{ k \in \{1, \dots, n\} : \mathbb{I}^j(\mathcal{M}_k^j(\mathbf{p}^j)) = 1 \right\}$$

**Optimal Approach** Finding the *imitation threshold* is a causal problem; The gold threshold can be achieved by performing the counterfactual experiment (Pearl, 2009): For each concept $Z^j$, create $k$ training datasets $\{\mathcal{D}_1^j, \mathcal{D}_2^j, \dots, \mathcal{D}_k^j, \dots\}$, and train a model $\mathcal{M}_k^j$ on each dataset $\mathcal{D}_k^j$. Once we find a model, $\mathcal{M}_k^j$ which is able to generate images where the concept $Z^j$ is recognizable, but $\mathcal{M}_{k-1}^j$ cannot, we deem $k$ as the *imitation threshold* for that concept. [2] *However, due to the high costs of training even one text-to-image model (Bastian, 2022), this approach is impractical.*

**MIMETIC[2]** Instead, we propose a tractable approach which efficiently estimates the imitation threshold while relying on certain assumptions (discussed at the end of this section). The key idea is to use observational data instead of training multiple models with a different number of images of a concept. This idea is a common approach to answer causal questions, when performing interventions is expensive, or unethical, inter alia, (Pearl, 2009; Lesci et al., 2024; Lyu et al., 2024). Concretely, we collect several concepts from the same domain (e.g., art styles) with varying image counts in the training dataset $\mathcal{D}$ of a pretrained model $\mathcal{M}$. Then, we identify the count where the model $\mathcal{M}$ starts imitating concepts, and deem this count to be the *imitation threshold* for that domain. **Note that, this threshold is domain specific and not concept specific**, where a *domain* is defined as the abstract set that contains the specific concept we want to measure imitation threshold for, e.g., if concept refers to Van Gogh's art style, then its domain would be art styles. **Therefore in this setup, MIMETIC[2] can estimate the imitation threshold for the domain of art styles, but it cannot estimate it for Van Gogh's art style specifically.**

**Assumptions** To estimate the imitation threshold using observational data, we make three assumptions. These are standard assumptions in the field, which are necessary for answering such complex questions. Crucially, we are able to empirically validate all of our assumptions, and find they hold for our datasets and models (Appendix C). First, we assume distributional invariance between the images of all concepts in a domain. Under this assumption, measuring the imitation score of a concept $Z^j$ for a counterfactual model $\mathcal{M}_{k'}$ that is trained with $k'$ images of $Z^j$ is equivalent to measuring the imitation score of another concept $Z^i$ that currently has $k'$ images in the already trained model $\mathcal{M}$

$$\text{Imitation Score}(\mathcal{M}_{k'}^j(Z^j)) \approx \text{Imitation Score}(\mathcal{M}_{k'}^i(Z^i))$$

This assumption allows us to use observational data to estimate the imitation threshold without training models from scratch.

Second, we assume that there are no confounders between the imitation score and the image count of a concept, i.e, the imitation score of a concept is not affected by the presence of other concepts in the training data. Since there are visual similarities between concepts, it might seem that such concepts might influence

---

[1]Prompts $\mathbf{p}$ are usually different from the training data captions, $\mathbf{y}$.

[2]In an ideal world, we can train $\mathcal{O}(log(k))$ models, with an estimated cost of \$10M if $k = 100,000$ (see Appendix O).

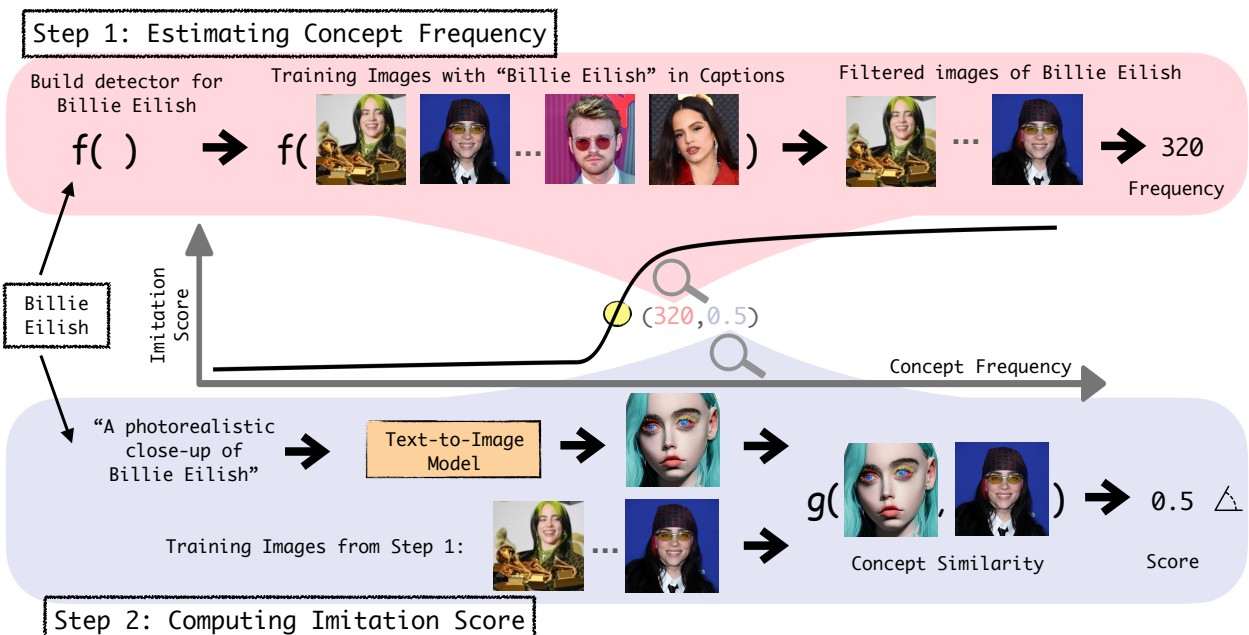

Figure 2: Overview of MIMETIC$^2$'s methodology to estimate the *imitation threshold*. In Step 1, we estimate the frequency of each concept (belonging to a domain) in the pretraining data by obtaining the images that contain the concept of interest. In Step 2, we use the filtered images of each concept (obtained in Step 1) and compare them to the generated images to measure imitation (using $g$ that receives training and generated images). We repeat this process for each concept to generate the imitation score graph, and then determine the *imitation threshold* with a change detection algorithm.

the imitation score of a concept – however, our discriminator models are able to distinguish between concepts with high accuracy, alleviating the concern for such a phenomenon.

Third, we assume each image of a concept contributes equally to the learning of the concept. We further discuss these assumptions and provide empirical evidence they hold for the datasets we experiment with in Appendix C.

# 4 Proposed Methodology: MIMETIC$^2$

We illustrate our proposed methodology in Figure 2. At a high level, for a specific domain (e.g., human faces), MIMETIC$^2$ estimates the frequency of each concept in the pretraining data (§4.1) and estimates the model's ability to imitate it (§4.2). We then sort the concepts based on their frequencies, and find the imitation threshold using a change detection algorithm (§4.3).

## 4.1 Computing Concept Frequencies

**Challenges** Determining a concept's frequency in a multimodal dataset can be achieved by employing a high-quality classifier for that concept over every image and counting the number of detected images. However, given the scale of modern datasets with billions of images, this approach is expensive and prone to classification errors.

**Estimating Concept Frequency**

We first make a simplifying assumption that a concept is present only if an image's caption mentions it. We empirically test the validity of this assumption and find it to be accurate (Appendix D). However, this is not sufficient because concepts often do not appear in the corresponding images, even when they are mentioned in the captions. For instance, Figure 3 showcases images whose captions contain "Mary Lee Pfeiffer", but such images do not always include her. On average, in our experiments we find that concepts occur only in 60% of the images whose caption mentions the concept.

To address this problem, we start by retrieving all images whose caption mentions the concept of interest and then further filter out the images that do not contain the concept, as detected by a classifier (we build a classifier that indicates the presence of a particular concept in the image). We retrieve these images using WIMBD (Elazar et al., 2024), a search tool based on a reverse index that efficiently finds documents (captions) in a dataset containing the search query (concept). To build a classifier for each concept we construct a set of high quality reference images. For example, a set of images with only the face of a single person (e.g., Brad Pitt). We collect these images automatically using a search engine (we use images from a search engine because several of the concepts in our experiments have a low number of images that cannot be used to build a classifier), followed by a manual verification to vet the images (see Appendix E for details). These images are used as gold reference for automatic detection of these concepts in the images. We collect up to ten reference images per concept.

To classify whether a candidate image contains the concept of interest or not, we embed the candidate image and the concept's reference images using an image encoder (discussed in §4.2) and measure the similarity between the embeddings. If the similarity between a candidate image and any of the reference images is above some threshold, we consider that candidate image to contain the concept of interest.

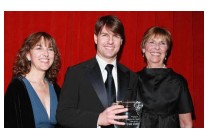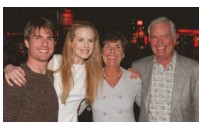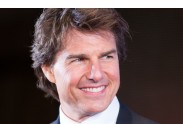

Figure 3: LAION captions that mention 'Mary Lee Pfeiffer', the mother of Tom Cruise. She is not always present in the images (the rightmost image).

This threshold is established by measuring the similarity between images of the same concepts and images of different concepts which maximizes the true positives, and minimizes false positives. We provide additional details on determining the thresholds per domain in Appendices F and G.

Finally, we run the classifier on all candidate images whose caption mentions the concept, and take those that are classified as positive. For each concept, we retrieve up to 100K candidate images from the pretraining dataset. We use the ratio of positive predictions to the total number of retrieved candidate images, and multiply it by the total caption count of a concept in the dataset and use that as the concept's frequency estimate. For concepts with less than 100K candidate images, we count the number of images that are positively classified. Note that some URLs from the pretraining datasets we use are dead, a common phenomenon for URL based datasets ("link rot" (Carlini et al., 2024; Lakic et al., 2023)). On average, we successfully retrieved 74% of the candidate images.

## 4.2 Computing the Imitation Score

**Challenges** Computing the imitation score entails determining how similar a concept is in a generated image compared to its source images from the training data. Several approaches were proposed to accomplish this task, such as FID and CLIPScore (Salimans et al., 2016; Heusel et al., 2017; Sajjadi et al., 2018; Hessel et al., 2021; Podell et al., 2024). To measure similarity, these approaches compute the similarity between the distributions of the embeddings of the generated and training images of a concept. The embeddings are obtained using image embedders like Inception model in case of FID and CLIP in case of CLIPScore. These image embedders often perform reasonably well in measuring similarity between images of common objects which constitutes most of their training data. However, they cannot reliably measure the similarity between two very similar concepts like the faces of two individuals or art style of two artists (Somepalli et al., 2024; Hessel et al., 2021; Ahmadi & Agrawal, 2024; Jayasumana et al., 2024). Therefore, MIMETIC$^2$ uses domain-specific image embedders to measure similarity between two concepts from that domain.

We embed the generated and real images of concepts using these embedders and use similarity to measure a continuous imitation score. To establish a threshold, we use a change detection algorithm (described next; §4.3). At a high level, given a sequence of imitation scores, the algorithm finds the threshold on the scores above which the score is considered a (binarized) imitation of the said concept. As we are interested in finding a (binary) threshold, we binarize the score, which we find to correspond to the human perception imitation of a concept. To verify the validity of the binarized scores we conduct human subject experiments and find high correlation between human perception of imitation and the binarized threshold (Section 6). We leave to future work the investigation of the continuous imitation score beyond the threshold.

**Estimating Imitation Score** We use a face embedding model (Deng et al., 2022) for measuring face similarity and an art style embedding model (Somepalli et al., 2024) for measuring art style similarity. We find that even the specific choice of these models is crucial; for instance, in early experiments we used Facenet (Schroff et al., 2015), and observe it struggles to distinguish between individuals of certain demographics, causing drastic differences in the imitation scores between demographics. We provide more details on these early experiments in Appendix M, and show that our final choice of embedding models work well on different demographics.

To measure the imitation score we embed the generated images and filtered training images of a concept (obtained in §4.1) using the domain specific image embedder, and report the imitation score as the average imitation between all generated and top-10 most similar training images (cosine similarity between embeddings of generated and training images). To ensure that the automatic measure of similarity correlates with human perception, we also conduct experiments with human subjects and measure the correlation between the similarities obtained automatically and in the human subject experiments. We find this correlation to be high for both domains we experiment with (§6). We also do human subject experiments to verify the correctness of the binarized threshold and find this correlation to be high as well (§6).

### 4.3 Detecting the Imitation Threshold

After computing the frequencies and the imitation scores for each concept, we sort them in an ascending order of their image frequencies. This generates a sequence of points, each of which is a pair of image frequency and the imitation score of a concept. We apply a standard change detection algorithm, PELT (Killick et al., 2012), to find the image frequency where the imitation score significantly changes. Change detection is a classic statistical approach to find the points where the mean value of a sequence (e.g. stock value or network traffic) changes significantly. We choose PELT because of its linear time complexity in computing the change point (van den Burg & Williams, 2022). We choose the first change point as the imitation threshold (see Appendix J for details about all change points). The application of change detection assumes that increasing the image counts beyond a certain threshold leads to a large jump in the imitation scores, and we find this assumption to be accurate in our experiments (Figure 5a).

To obtain error bounds for our thresholds we perform a permutation test by sampling a subset of concepts per domain and dataset, and compute the threshold for the sampled set. We repeat this 1,000 times and report the mean and standard error of the thresholds.

## 5 Experimental Setup

Table 1: Domains, datasets, pretraining data, and models we use.

| Domain | Dataset | Pretraining Data | Model |
|---|---|---|---|
| Human Faces 🧒 | Celebrities, Politicians | LAION400M | LDM |
| Human Faces 🧒 | Celebrities, Politicians | LAION2B-en | SD1.1, SD1.5 |
| Human Faces 🧒 | Celebrities, Politicians | LAION5B | SD2.1 |
| Art Style 🖼 | Classical, Modern | LAION400M | LDM |
| Art Style 🖼 | Classical, Modern | LAION2B-en | SD1.1, SD1.5 |
| Art Style 🖼 | Classical, Modern | LAION5B | SD2.1 |

Table 2: Prompts used to generate images of human faces and art styles.

| Human faces 🧒 | Art style 🖼 |
|---|---|
| A photorealistic close-up photograph of X | A painting in the style of X |
| High-resolution close-up image of X | An artwork in the style of X |
| Close-up headshot of X | A sketch in the style of X |
| X's facial close-up | A fine art piece in the style of X |
| X's face portrait | An illustration in the style of X |

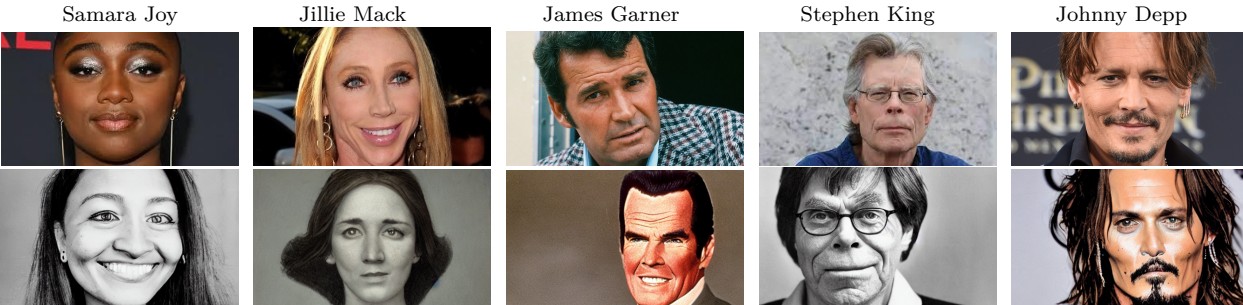

| Samara Joy | Jillie Mack | James Garner | Stephen King | Johnny Depp |

Figure 4: Examples of real celebrity images (top) and generated images from a text-to-image model (bottom) with increasing image counts from left to right (`3`, `273`, `3K`, `10K`, and `90K`, respectively). The prompt is "a photorealistic close-up image of {name}".

Table 3: The mean and std. error of the *imitation thresholds* for the models and domains we experiment with.

| | | Human Faces 👧 | | Art Style 🖼 | |
| --- | --- | --- | --- | --- | --- |
| **Pretraining** | **Model** | **Celebrities** | **Politicians** | **Classical** | **Modern** |
| LAION-400M | LDM | $626 \pm 3$ | $449 \pm 2$ | $416 \pm 3$ | $412 \pm 2$ |
| LAION2B-en | SD1.1 | $399 \pm 3$ | $284 \pm 3$ | $304 \pm 2$ | $208 \pm 1$ |
| | SD1.5 | $371 \pm 1$ | $302 \pm 4$ | $302 \pm 2$ | $212 \pm 1$ |
| LAION-5B | SD2.1 | $617 \pm 5$ | $385 \pm 5$ | $330 \pm 4$ | $292 \pm 3$ |

**Text-to-image Models and Training Data** We use Stable Diffusion (SD) as the text-to-image models (Rombach et al., 2022). We use them because both the models and their training datasets are open-sourced. Specifically, we use SD1.1 and SD1.5 that were trained on LAION2B-en, a 2.3 billion image-caption pairs dataset, filtered to contain only English captions and we use SD2.1 that was trained on LAION-5B, a 5.85 billion image-text pairs dataset that includes captions in any language (Schuhmann et al., 2022). Finally, we also use a latent diffusion model (LDM) trained on LAION-400M (Schuhmann et al., 2021), a 400M image-text pairs dataset.

**Domains and Concepts** We experiment with two domains – *art styles* 🖼 and *human faces* 👧 that are highly important for privacy and copyright considerations of text-to-image models. Figures 1 and 4 show examples of real and generated images of art styles and human faces. We collect two sets of concepts for each domain. For art styles we collect classical and modern art styles and for human faces, we collect celebrities and politicians. It is important to note that each artist's style and each human face is considered as a separate concept. Although in the real world art style of some artists might be similar, the classifier models we use to measure imitation is trained well to distinguish between art styles from different artists (Somepalli et al., 2024). These sets we experiment with are independent (i.e., have no common concept) and are therefore useful to test the robustness of the thresholds (§6). Each set has 400 concepts that cover a wide frequency range in the pretraining datasets. Appendix N provides details of the sources used to collect the concepts and the sampling procedure. Table 1 summarizes the domains, sets used for each domain, the models and their pretraining datasets we experiment with.

**Image Generation** We generate images for each domain by prompting models with five prompts (Table 2). We design domain-specific prompts that encourage the desired concept to occupy a large part of the generated image, which simplifies the imitation score measurement. We also ensure that these prompts are distinct from the captions used in the pretraining dataset to minimize direct reproduction of training images (as noted by (Somepalli et al., 2023b)). We generate 200 images per concept using different random seeds for each prompt, a total of 1,000 images per concept.

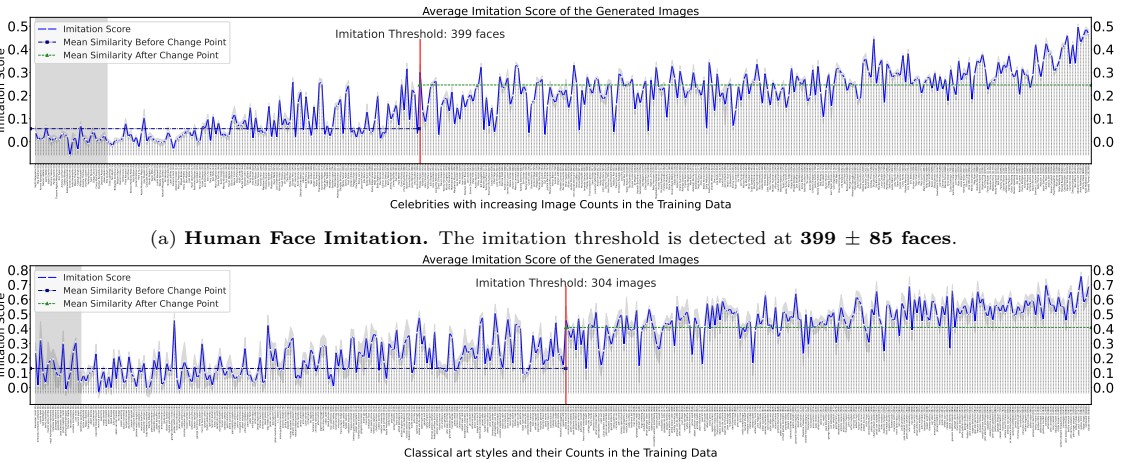

(a) **Human Face Imitation.** The imitation threshold is detected at **399 $\pm$ 85 faces**.

(b) **Art Style Imitation.** The imitation threshold is detected at **304 $\pm$ 56 images**.

Figure 5: **Human Face** and **Art Style** imitation graphs for SD1.1 using the *Celebrities* and *Classical art style* sets. The x-axis is the image frequencies and the y-axis is the imitation score averaged over the five prompts. Concepts with zero image frequencies are shaded in light gray. We show the mean imitation score and its variance over the five image generation prompts for each concept. The red vertical line indicates the imitation threshold found by the change detection algorithm, and the horizontal green line represents the average imitation scores before and after the threshold.

# 6 Results: The *Imitation Threshold*

We use MIMETIC$^2$ to estimate the imitation threshold for each model-data pair, and present the results in Table 3. The imitation thresholds for SD1.1 on celebrities and politicians are 399 $\pm$ 85 and 284 $\pm$ 87 respectively, and 304 $\pm$ 56 and 208 $\pm$ 26 for classical and modern art styles, respectively. Interestingly, SD1.1 and SD1.5 have almost identical thresholds across the four datasets. Notably, both SD1.1 and SD1.5 are trained on LAION2B-en. The imitation thresholds for SD2.1, which is trained on LAION-5B (a superset of LAION-2B) are higher than the thresholds for SD1.1 and SD1.5. We hypothesize that the difference in performance of SD2.1 and SD1.5 is due to the difference in their text encoders (note: differences in performance between SD1.5 and SD2.1 were also reported by several users on online forums (O'Connor, 2022)). Finally, the thresholds for all four datasets are slightly higher for LDM model compared to all the SD models. We hypothesize that this might happen due to the smaller training data size of 400M pairs and leave further investigation for future work.

We also present the plots of the imitation scores as a function of the image frequencies of the concepts for the four datasets. Figures 5a and 5b show the imitation graphs of celebrities and classical art styles, respectively for SD1.1. In Figure 5a, we observe that the imitation scores for individuals with low image frequencies are close to 0 (left side), and increase as the image frequencies increase (towards the right side). The highest similarity occurs for individuals in the rightmost region of the plot. We also observe a low variance in the imitation scores across prompts for the same concept (the mean of the variance is 0.0003 with a standard deviation of 0.0005 across concepts), and note that this variance is independent of the concept frequencies – indicating that the performance of the face embedding model does not depend on the popularity of the individual. Similarly, in Figure 5b, we observe that imitation scores for art styles with low image frequencies are low (close to 0.2 on the left side), and increase as the image frequencies increase (towards the right side). The highest similarity occurs for the artists in the rightmost region of the plot. We also notice a low variance across the generation prompts, and the variance does not depend on the popularity of the artist (the mean of the variance is 0.003 with a standard deviation of 0.003). We showcase the imitation graphs for all other models in Appendix K, which follow similar trends.

**Human Perception Evaluation** To determine if the automatic measure of similarity between generated and training images matches human perception, we conduct experiments with human subjects. We ask participants to rate generated images on the Likert scale (Likert, 1932) of 1-5 based on their similarity to real images of celebrities. To avoid any confirmation bias, the participants were not informed of the research objective of this work.

For human face imitation, we conduct this study with 15 participants who were asked to rate 100 (randomly selected) generated images for a set of 40 celebrities on the scale of 1-5. To determine the accuracy of the imitation threshold estimated by MIMETIC$^2$, we randomly select the celebrities such that half of them have image frequencies below the threshold and the other half above it We measure the Spearman's correlation (Zar, 2005) between the imitation scores computed by the model and the ratings provided by the participants. Due to the variance in perception, we normalize the ratings for each participants. The Spearman correlation between the similarity scores provided by participants and the imitation scores is **0.85, signifying a high quality imitation estimator.** We also measure the agreement between the imitation threshold that MIMETIC$^2$ estimates and the threshold that humans perceive. For this purpose, we convert the human ratings to binary values and treat it as the ground truth (any rating of 3 or higher is treated as 1 and ratings less than 3 are treated as 0, indicating successful and unsuccessful imitation, respectively). MIMETIC$^2$ predictions is also binarized, any celebrity with frequency higher than the imitation threshold is treated as 1, otherwise 0. To measure the agreement, we compute the element-wise dot product between these two sets. We find the agreement to be 82.5%, signifying a high degree of agreement for MIMETIC$^2$'s automatically computed threshold.

For art style imitation, we conduct this study with an art expert due to the complexity of detecting art styles. The participant was asked to rate five generated images for 20 art styles, half of which were below the imitation threshold and the other half, above the threshold. We find the Spearman correlation between the two quantities to be **0.91** – demonstrating that our imitation scores are highly correlated with an artist's perception of style similarity. Similar to the previous case, we measure the agreement of the imitation threshold, which we find to be 95% – signifying a high degree of agreement for MIMETIC$^2$'s computed threshold.

**Evaluation of the Imitation Thresholds** Evaluating the accuracy of the real imitation thresholds is expensive, as it requires conducting the counterfactual experiment described in Section 3 (training $\mathcal{O}(\log m)$ models with different number of images of each concept). To make the evaluation tractable, we finetune a pretrained text-to-image model with concepts that the model has not seen during pretraining and find the number of images required for the model to imitate them. We find that all the concepts require about 250 images for imitation when the models are finetuned on them (further details in Appendix H), which is very close to the imitation threshold MIMETIC$^2$ finds for this pretrained model without finetuning (234 images). Based on these results, we conclude that the imitation thresholds estimated by MIMETIC$^2$ are accurate.

**Discussion** Overall, we observe that the imitation thresholds are similar across the different image generation prompts, but are domain and model dependent. Importantly, the thresholds computed by MIMETIC$^2$ have a high degree of agreement with human subjects and are supported by the finetuning experiments.

We find that celebrities have a higher imitation threshold than politicians for all models. We hypothesize this happens due to inherent differences in the data distribution between these two sets, which makes it harder to learn the concept of celebrities than politicians. To test this hypothesis, we compute the average number of training images that has *a single person* for concepts with less than 1,000 images. We find that politicians have about twice the number of single person images compared to celebrities. As such, images that have a single person makes imitation easier for the model (the model can easily associate the face with the name in the caption), thus lowering the imitation threshold for politicians.

We also note the presence of several outliers in both plots, which can be categorized into two types: (1) concepts whose image frequencies are lower than the imitation threshold, but their imitation scores are considerably high; and (2) concepts whose image frequencies are higher than the imitation threshold, but their imitation scores are low. From a privacy perspective, the first kind of outliers are more concerning than the second ones, since the imitation threshold should act as a privacy threshold. When used as a privacy guideline, it is less worrisome if a concept with a concept frequency higher than the threshold is not imitated by the model (false positive), but it would be a privacy violation if a model can imitate a concept with frequency lower than the threshold (false negative). Upon further analysis, we find that for all of the false negative outliers (with the privacy concern), the concepts' true frequencies are much higher than our estimates. This happens primarily due to *naming aliases* (a person known by multiple names) that MIMETIC$^2$ does not account for – thereby alleviating the privacy violation concerns (see §7 for a detailed analysis of such outliers).

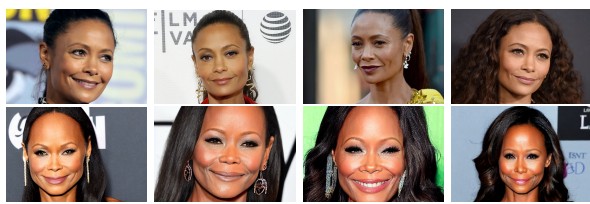 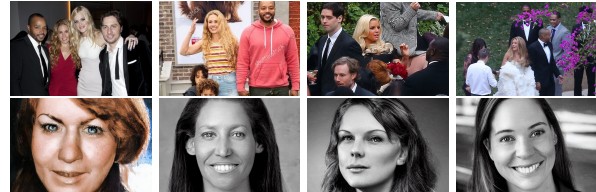

(a) **Outlier Kind 1.** *Thandiwe Newton* is also aliased as *Thandie Newton*. Since MIMETIC² only collects images whose caption mentions *Thandiwe Newton*, this leads to underestimation of image counts.

(b) **Outlier Kind 2.** Most of the images whose captions mention *Cacee Cobb* have multiple people in them, only 6 images have her as the only person, leading to a low imitation score in generated images.

Figure 6: Examples of the two kinds of outliers. The top and bottom rows show the real and SD1.1 generated images respectively. Images were generated using the prompt: "a photorealistic close-up image of *name*."

Finally, we notice that the range of the imitation scores of different domains have different y-axis scales. This is due to the difference in embedding models used in both cases. The face embedding model can distinguish between two faces much better than the art style model can distinguish between two styles (see Appendices F and G), and therefore the scores for the concepts on the left side of the imitation threshold is around 0 for face imitation and 0.2 for style imitation. However, the absolute values on the y-axis do not matter for estimating the imitation threshold as long as the trend is similar, which is the case for both domains.

## 7 Analysis: Investigating Outliers

The imitation score plots in the previous section, while showcasing a clear trend, have several outliers. In this section, we analyze such outliers and present examples in Figure 6 (more outliers can be found in Figures 36 and 37 in the Appendix).

**Low Image Counts and High Imitation Scores** Figure 6a shows an example of such a case: *Thandiwe Newton*'s image count is 172 in LAION2B-en, lower than the *imitation threshold* for celebrities: 364. However, her imitation score of 0.26 is much higher than those of neighboring celebrities with similar image counts (with scores of 0.01 and 0.04). Further investigation reveals that Thandiwe Newton is also known as *Thandie Newton*. Since this alias may also be used to describe her in captions, we underestimate her image counts. We repeat the process for estimating the image counts with the new alias, and find that *Thandie Newton* appears in 12,177 images, bringing the cumulative image count to 12,349, which significantly surpasses the established imitation threshold. The two aliases, whose total image count is considerably higher than the imitation threshold, differ by only a single letter and are similarly represented by the model's text encoder (cosine similarity of 0.96), which explains the high imitation score. We find that most of the celebrities who had a high imitation score while having a low concept frequency are also known by other names which lead to underestimating their image counts. For example, *Belle Delphine* (394 images) also goes by *Mary Belle* (310 images), for a total of 704, and *DJ Kool Herc* (492 images) also goes by *Kool Herc* (269 images), for a total of 761. The aliases explanation also explains the outliers in art style imitation. For instance, artist *Gustav Adolf Mossa* (19 images) also goes by just *Mossa* (15,850 images). See Figures 38 and 39 in the appendix for the real and generated images in style of these artists.

**High Image Counts and Low Imitation Scores** Several celebrities have higher image counts than the imitation threshold, but low imitation scores. Unlike the previous case, we were unable to find a common cause that explains all these outliers. However, we find explanations for specific cases. For example, a staggering proportion of the training images for such celebrities have multiple people in them (20% of the outliers of this kind could be explained with this phenomenon). Out of the 706 total images of *Cacee Cobb*, only 6 images have her as the only person in the image (see Figure 6b). Similarly, out of 1,296 total images of *Sofia Hellqvist*, only 67 images have her as the only person and out of the 472 total images of *Charli D' Amelio*, only 82 images have her as the only person. We hypothesize that having multiple concepts in an image impedes the proper mapping of the concept's text embedding to its image embedding, which can explain the low imitation score for these concepts. We leave examining this to future work.

## 8 Discussion and Limitations

In this work we attribute the imitation of a concept by a model to its frequency in the training data. However, several other factors like image resolution, image diversity, alignment between images and their captions, etc., may affect imitation. Our work is the first to tackle such important question, and as such, we tackle one of the more salient features that contributes to imitation - the concept's frequency in the training data. In the future, we plan to incorporate other features to improve the accuracy of the imitation threshold. While we hope our work could be used for copyright and privacy claims, our results should be put in content of the assumptions we make (which we find to empirically hold). We further discuss the assumptions we make, and their empirical validity in more detail in Appendix B. Note that, MIMETIC$^2$ finds the imitation threshold for the pretraining regime. Different training regimes (such as finetuning on additional data, for multiple epochs) may also affect the imitation threshold, and we hope to investigate such scenarios in future work.

In this work we are interested in finding the estimation threshold of text-to-image models, studying how many images of a given concept a model has to be trained on to be able to generate images of such concepts. We build a system to find such threshold for a given model, dataset, and domain, and show how to estimate such threshold efficiently. We test our method on various domains, datasets, models, and pretraining datasets, and show that the thresholds we estimate are in the same ball park, while exploring some of the variance in our analysis section. Notably, we cannot apriori know how such threshold would generalize to other models, e.g., larger models or models trained on larger datasets that have a different parameter-to-generalization ability. Additional work is needed to study such cases. However, our main contribution in this work is the method (MIMETIC$^2$) that allows to study such threshold and can be applied in future works on additional models.

## 9 Conclusions

Text-to-image models can imitate concepts from their training images (Somepalli et al., 2023a; Carlini et al., 2023a; Somepalli et al., 2023b) This behavior is crucial for learning, but it can also be concerning when such training datasets include copyrighted and private images. Imitating such images would be grounds for violation of copyright and privacy laws. In this work, we seek to find the number of instances of a concept that a text-to-image model has to be trained on in order to imitate it – the *imitation threshold*. We posit this as a new problem and propose an efficient method, MIMETIC$^2$, for estimating such threshold. Our method uses pretrained models to estimate this threshold for human faces and art styles imitation using four text-to-image models trained on three different pretraining datasets. We find the imitation threshold of these models to be in the range of 200-700 images depending on the setup. By estimating the imitation threshold, we provide insights on successful concept imitation based on their training frequencies. Our results have striking implications on both the text-to-image models users and developers. These thresholds can inform text-to-image model developers what concepts are in risk of being imitated, and on the other hand, serve as a basis for copyright and privacy complaints. Finally, MIMETIC$^2$ also provides insights into the generalization capabilities of text-to-image models.

## 10 Ethics Statement

This work is targeted towards understanding and mitigating the negative social impacts of widely available text-to-image models. To minimize the impact of our study on potential copyright infringement and privacy violations, we used existing text-to-image models and generation techniques. To strike a balance between reproducibility and ethics, we release the code used in this work, but not the real and synthetic images used (we also provide detailed documentation of our data curation process in the code to ensure transparency).

**Human Subject Experiments**  For the human subject experiments performed in this work, we followed the university guideline. The study was IRB approved by the university and each participant was made aware of the goals and risks of the study. In accordance with local minimum wage requirements, we compensated each participant with Amazon gift cards of $25/hour for their time.

## 11   Reproducibility Statement

We have published all the code used for this work at https://github.com/vsahil/MIMETIC-2.git and included the `README.md` file that provides instructions to execute each step of the algorithm. We provide our code in the supplementary material. The code includes a detailed `README.md` file that provides instructions to execute each step of the algorithm. We will open-source all the code upon publication of the paper. We do not provide the original and generated images of individuals and art styles for copyright and privacy concerns.

## ACKNOWLEDGMENTS

We would like to thank Shauli Ravfogel, Moni Shahar, Amir Feder, Shruti Joshi, and Soumye Singhal for discussion and providing feedback on the draft. In addition, we would like to thank the annotators of our human experiments: Miki Apfelbaum, and the Angle icon by Royyan Wijaya from the Noun Project (CC BY 3.0). A part of this work was supported using Amazon Research Award (ARA) to Chirag Shah and NSF Grant Numbers IIS-2106937 and IIS-2148367 awarded to Jeff Bilmes.

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

## A  Additional Related Work

**Imitation in Text-to-Image Models:**  Somepalli et al. (2023a); Carlini et al. (2023a) demonstrated that diffusion models can memorize and imitate duplicate images from their training data (they use 'replication' to refer to this phenomenon). Casper et al. (2023) corroborated the evidence by showing that these models imitated art styles of 70 artists with high accuracy (as classified by a CLIP model) when prompted to generated images in their styles (a group of artists also sued Stability AI claiming that their widely-used text-to-image models imitated their art style, violating copyright laws Saveri & Butterick (2023)). However, these works did not study how much repetition of a concept's images would lead the model to imitate them. Studying this relation is important as it serves to guide institutions training these models who want to comply with copyright and privacy laws.

**Mitigation of Imitation in Text-to-Image Models:**  Several works proposed to mitigate the negative impacts of text-to-image models. Shan et al. (2023) proposed GLAZE that adds imperceptible noise to the art works such that diffusion models are unable to imitate artist styles. A similar approach was proposed to hinder learning human faces (Shan et al., 2020). Wang et al. (2024) proposed adding noise to training images, which can be used to detect if a model has been trained on those images. Lu et al. (2024) propose pushing the generated images away from the distribution of training images to minimize mitigation. Kumari et al. (2023); Gandikota et al. (2024) proposed algorithms to remove specific styles, explicit content, and other copyrighted material learned by text-to-image models. On a related note, Xie et al. (2024) proposed Diffusion-ReTrac that finds training images that most influenced a generated image, and thereby provide a fair attribution to training data contributors.

## B  Considerations When Using Our Imitation Thresholds

Our work estimates the imitation thresholds under certain assumptions: distributional invariance between images of different concepts and absence of confounders between the imitation score and image count of a concept. While these assumptions are reasonable (as we discuss in Section 3 and empirically demonstrate in Appendix C.1) and have been also made in several prior works, they might limit the direct transfer of the imitation thresholds we report to other settings.

Our results also depend on the specific concepts we sampled for each domain, the specific domain embedder we use, and the specific change detection algorithm we use. Changing any of these might alter the imitation thresholds we report. We have performed experiments under several changes to these factors and found our claim to hold. However, we still do not claim our results would generalize to all model/data pairs. Specifically, these are some of the question to consider using our imitation thresholds in a different situation:

1. Is the setting being considered uses the same text-to-image model as we use in our experiments?

2. Is the setting being considered uses the same pretraining dataset as we use in our experiments?

3. Does the setting being considered uses the imitation threshold we report for human face or art style imitation?

4. If our experimental procedure is repeated for a different setting, are the repeated experiments using a similar set of concepts in a domain as we use?

5. If our experimental procedure is repeated for a different setting, is our concept detector or a similar high quality concept detector is being used?

Finally, while we hope our work could be used for copyright and privacy claims, our results should be put in context of the assumptions we make in (Section 3)

# C   Verification of the Assumptions

We empirically test the validity of the three assumptions we make in Section 3, and find they all hold in practice. To test the invariance assumption we calculate the imitation scores difference of all concepts pairs, whose image counts differ by less than 10 images, for all concepts. We find that the imitation scores difference for such pairs, on average, is 0.003 (averaged across such pairs and averaged across our datasets) (Appendix C.1), attesting to the validity of this assumption. Testing the lack of confounders between imitation score and image count, we finetune a text-to-image model on images of concepts that the model has not seen during pretraining (e.g., politicians that became popular after the training data of the model was curated), and measure the imitation threshold for these concepts Appendix C.2. We argue that if the thresholds found in the finetuning experiment and by MIMETIC$^2$ are close, then the assumption about lack of confounders is valid (assuming there is no other confounder affecting all these concepts). We indeed find that the imitation thresholds found in the finetuning experiment are very close to the thresholds found by MIMETIC$^2$ without finetuning the model ($\pm6\%$), attesting to the validity of this assumption (Section 6). Lastly, we test the assumption that each image of a concept contributes equally to the learning of a concept, which is commonplace in the sample complexity literature (Valiant, 1984; Wang et al., 2017). We finetune a text-to-image model with a set of randomly sampled images of a concept multiple times. We report the variance in the imitation scores across these finetuning experiments and find the imitation scores variance to be small ($\pm0.2\%$ of the imitation score), validating this assumption (see more details in Appendix C.3).

To summarize, we empirically verify all the assumptions we make in this work, and find them to hold in practice.

## C.1   Validity of the invariance assumption

Table 4: Average difference in the imitation scores for concepts whose image counts differ by less than 10. The difference in the imitation scores are close to 0, empirically validating the distribution invariance assumption.

| Domain | Dataset | Avg. difference in imitation score |
|---|---|---|
| **Human Faces** 🧑 | Celebrities | 0.0007 |
| **Human Faces** 👦 | Politicians | 0.0023 |
| **Art Style** 🖼 | Classical Art Style | -0.0088 |
| **Art Style** 🖼 | Modern Art Style | -0.0013 |

In this section, we empirically test the statistical validity of the assumptions we make in Section 3. The first assumption states that there is a distributional invariance across concepts. If this is true, then the imitation scores of two concepts (from the same domain) whose image counts are similar, should also be similar to each other. To test whether this is empirically true for the domains we experiment with, we measure the difference in the imitation scores for concepts whose image counts differ by less than 10 images and report the difference averaged over all such pairs. Table 4 shows that the average difference in the imitation scores for all the pairs, for all the datasets we experiment with, is very close to 0 (also see Figure 7 for the distribution of the differences in the score). This provides empirical validation of the distribution invariance assumption.

## C.2   Validity of the assumption about absence of confounders

To test the absence of confounder, we conduct a finetuning experiment in which we finetuned a pre-trained text-to-image model (SD1.4) on images of concepts the model has not seen during pretraining. We do this by finding politicians who recently got popular, after the SD1.4 model was open-sourced. We finetune SD1.4 on increasing number of images of these politicians, starting with 50 images (much below the imitation threshold found in the non-finetuning setup) and going upto 800 images (above the imitation threshold found in the non-finetuning setup). During the finetuning process, the images of the concepts were mixed with other 10,000 images taken randomly from LAION-2B-en dataset. This was done in order to ensure that our finetuning setup closely resembles the original training setup of SD1.4 where images of a concept would be naturally interspersed with other images in the dataset. We finetune on the full dataset of 10,000 images for 1 epoch with a learning rate of $5e-5$. We use the following concepts in this experiment:

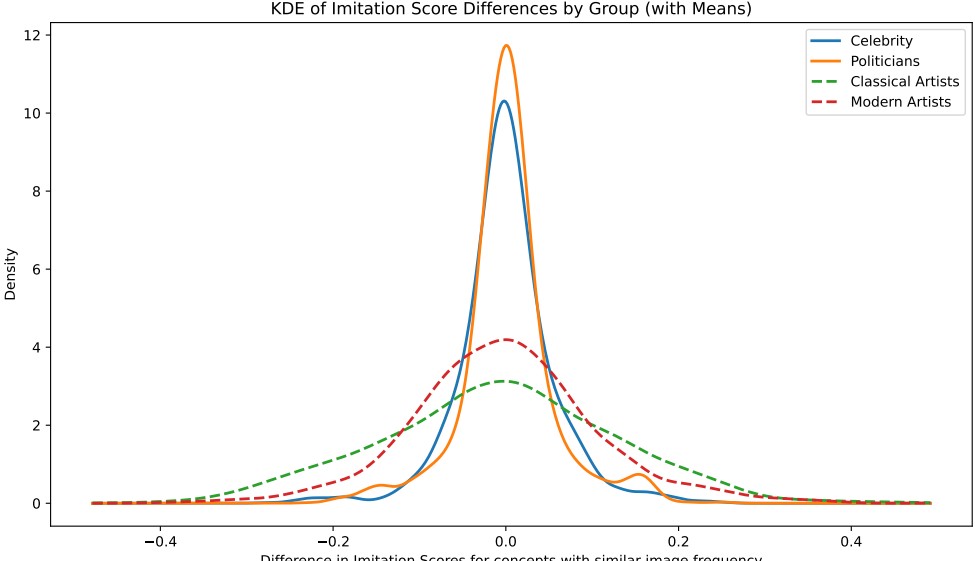

Figure 7: This figure shows the distribution of differences in imitation scores for concepts whose image counts differ by less than 10 images. We use KDE for this estimation. Since the differences are very small for all the four domains we experiment with, we can argue that our assumption about distributional invariance is valid.

Averie Bishop, Sophie Wilde, Javier Milei, Hashim Safi al-Din, Shyam Rangeela, Akshata Murty, Hana-Rawhiti Maipi-Clarke, Sébastien Lecornu, Vivek Ramaswamy, Raghav Chaddha.

For the SD1.4, the imitation threshold for the non-finetuning setup is found to be at 234 images (Figure 19) and the average imitation scores for the politicians on the right hand side of the threshold is **0.26**. We report the results in Table 5, which shows that the imitation score remains lower than **0.26** when 50, 100, 150, or 200 images are used to finetune the model. The imitation score reaches this value of 0.26, only when finetuned with 250 or more images. This imitation threshold of 250 in the finetuning experiment is very close to the imitation threshold MIMETIC$^2$ finds for SD1.4 without requiring to finetune the model (234 as shown in Figure 19).

Note that since in this experiment we are directly intervening on the image count of each concept, there is no effect of any confounder whatsoever even if they were originally present. And the closeness of the imitation threshold in the finetuning and the non-finetuning setup alludes to the absence of confounders, thus validating our assumption. Note that this finetuning experiment is also the optimal experiment for finding the imitation threshold, that we described in Section 3. Based on these results, we can also claim that imitation thresholds MIMETIC$^2$ finds are close to the ground-truth thresholds that the optimal experiment would find.

### C.3 Validity of the assumption about equal image contribution

We conduct another finetuning experiment to test the validity of our third assumption (which assumes that all images of a concept contribute equally) to its learning. In this experiment, we finetune a pre-trained SD1.4 model on images of concepts that the model has not seen during pre-training (these are the same concepts we used for the finetuning experiment in Appendix C.2). Specifically, in this experiment we sample 400 images of these politicians from a random pool of images 20 times, and finetune a SD1.4 model with each of the sampled set of images. For each of the 20 finetuned models, we then measure the imitation scores for these concepts, and report the mean and standard deviation in the scores (also see Figure 9 for the distribution of the imitation scores) Table 6 shows these values, and the low standard deviation in the imitation scores indicate that most images contribute equally to the learning of the concept, thus validating our assumption.

Table 5: The imitation scores of the concepts as a text-to-image model, SD1.4, is finetuned on increasing number of images of a concept that it has not seen during pretraining. We report imitation scores averaged over 10 such concepts (politicians). We notice that the imitation scores have values 0.26 or greater when the model is finetuned on 250 or more images of a politician. Critically, this is the average imitation score for politicians on the right hand side of the imitation threshold in Figure 19 (which is the imitation of politicians for SD1.4), indicating that 250 images or more are required for imitation in this experiment.

| # Finetuning Images | Imitation Score After Finetuning (Avg. over 10 politicians) |
|---|---|
| 50 | 0.07 |
| 100 | 0.16 |
| 150 | 0.19 |
| 200 | 0.21 |
| 250 | 0.26 |
| 300 | 0.27 |
| 400 | 0.29 |
| 500 | 0.31 |
| 600 | 0.32 |
| 700 | 0.36 |
| 800 | 0.34 |

Table 6: Mean and std. deviation of the imitation scores when we finetuned a pre-trained SD1.4 model on 400 images of five concepts. We sample the images used for finetuning from a pool of images of these concepts. We repeat this process 20 times, and measure the mean and std. deviation in the imitation scores. The low std. deviation in the imitation scores indicate that most images contribute equally to the learning of the concept, thus validating our assumption.

| Concept | Mean of Imitation Scores | Std. Deviation of Imitation Scores |
|---|---|---|
| Javier Milei | 0.36 | 0.02 |
| Hashim Safi al-Din | 0.38 | 0.03 |
| Shyam Rangeela | 0.31 | 0.03 |
| Akshata Murty | 0.30 | 0.03 |
| Hana-Rawhiti Maipi-Clarke | 0.30 | 0.02 |

## D  Caption Occurrence Assumption

For estimating the concept's counts in the pretraining dataset we make a simplifying assumption: a concept can be present in the image only if it is mentioned in a paired caption. While this assumption isn't true in general, we show that for the domains we experiment with, it mostly holds in practice.

Table 7: We estimate the percentage of the images that our approach misses due to the assumption that a concept only occurs if it is mentioned in the caption. The small percentage of the images we miss (rightmost column) shows that our assumption of counting the images where a concept is mentioned in the caption is empirically reasonable.

| Celebrity | Face Count in 100K images | Face Count in Images with Caption Mention | Missed Images (%) |
|---|---|---|---|
| Floyd Mayweather | 1 | 0 | 0.001% |
| Oprah Winfrey | 2 | 0 | 0.002% |
| Ronald Reagan | 6 | 3 | 0.003% |
| Ben Affleck | 0 | 0 | 0.0% |
| Anne Hathaway | 0 | 0 | 0.0% |
| Stephen King | 0 | 0 | 0.0% |
| Johnny Depp | 9 | 1 | 0.008% |
| Abraham Lincoln | 52 | 1 | 0.051% |
| Kate Middleton | 34 | 1 | 0.033% |
| Donald Trump | 16 | 0 | 0.016% |

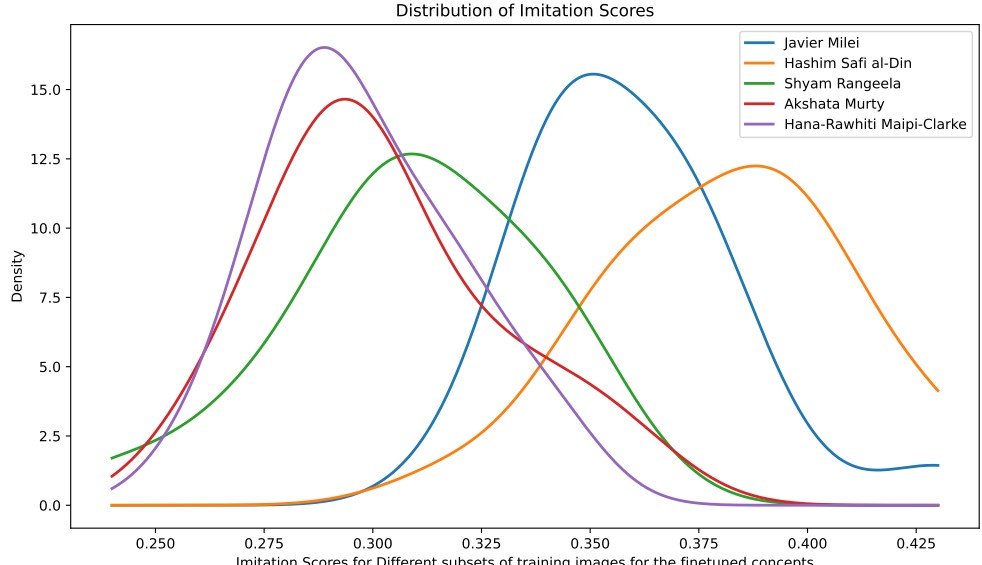

Figure 8: Distribution of imitation scores for the concepts we finetuned on. We use KDE for this estimation.

Figure 9: This figure shows the distribution of imitation scores for concepts we finetuned on using different subsets of training images. We use KDE for this estimation. Since the imitation scores are similar for different subsets of training images for all concepts (demonstrated by narrow curves), we can argue that our assumption of each image contributing equally to model learning is valid.

For this purpose, we download 100K random images from LAION2B-en, and run the face detection (used in Section 4) on all images, and count the faces of the ten most popular celebrities in our sampled set of celebrities. Out of the 100K random images, about 57K contain faces. For each celebrity, we compute the similarity between all the faces in the downloaded images and the faces in the reference images of these celebrities. If the similarity is above the threshold of 0.46, we consider that face to belong to the celebrity (this threshold is determined in Appendix F to distinguish if two images are of the same person or not). Table 7 shows the number of faces we found for each celebrity in the 100K random LAION images. We also show the face counts among these images whose captions mention the celebrity. We find that 1) the highest frequency an individual appears in an image without their name mentioned in the caption is 51 (*Abraham Lincoln* is mentioned once in the caption and he appears a total of 52 times), and 2) the highest percentage of image frequency that we miss is 0.051%, and 3) most of the other miss rates are much smaller (close to 0). Such low miss rates demonstrate that our assumption of counting images when a concept is mentioned in the caption is empirically reasonable.

We also note that this assumption would fail if we were computing image frequencies for concepts that are so widely common that one would not even mention them in a caption, for example, phone, shoes, or trees.

# E    Collection of Reference Images

## E.1    Collection of Reference Images for Human Faces Domain

The goal of collecting reference images is to use them to filter the images of the pretraining dataset. These images are treated as the gold standard reference images of a person and images collected from pretraining dataset are compared to these images. If the similarity is higher than a threshold then that image is considered to belong to that person (see Section 4 for details). We describe an automatic manner of collecting the reference images. The high level idea is to collect the images from Google Search and automatically select a subset of those images that are of the same concept (same person's face or same artist's art). Since this is a crucial part of the overall algorithm, we manually vet the reference images for all the concepts to ensure that they all contain the same concept.

**Collection of Reference Images for Human Face Imitation:** We collect reference images for celebrities and politicians using a three step process (also shown in Algorithm 1):

1. **Candidate set:** First, we retrieved the first hundred images by searching a person's name on Google Images. We used SerpAPI (SerpApi, 2024) as a wrapper perform the searches.

2. **Selecting from the candidate set:** Images retrieved from the internet are noisy and might not contain the person we are looking for. Therefore we filter images that contain the person from the candidate set of images. For this purpose, we use a face recognition model. We embed all the faces in the retrieved images using a face embedding model and measure the cosine similarity between each one of them. The goal is to search for a set of faces that belong to the same person and therefore will have a high cosine similarity to each other.

   One strategy is for the faces to form a graph where the vertices are the face embeddings and the edges connecting two embeddings have a weight equal to the cosine similarity between them, and we select a dense k-subgraph (Lanciano et al., 2024) from this graph. Selecting such a subgraph means finding a mutually homogeneous subset. We can find the vertices of this dense k-subgraph by cardinality-constrained submodular function minimization (Bilmes, 2022; Nagano et al., 2011) on a facility location function (Bilmes, 2022). We run this minimization and select a subset of images (at least of size ten) that has the highest average cosine similarity between each pair of images.

3. **Manual verification:** Selecting the faces with the highest average similarity is not enough. This is because in many cases the largest set of faces in the candidate set are not of the person we look for, but for someone closely associated with them, in which case, the selected images are of the other person. For example, all the selected faces for *Miguel Bezos* were actually of *Jeff Bezos*. Therefore, we manually verify all the selected faces for each person. In the situation where the selected faces are wrong, we manually collect the images for them, for example, for Miguel Bezos. We collect at least 5 reference images for all celebrities.

---

**Algorithm 1:** Collection of Reference Images for Human Face Imitation

**Input:** Person's name P
**Output:** Verified Set of Images of P
$images \leftarrow$ SerpAPI(P) ;                    ▷ Retrieve initial image set using SerpAPI
$candidateSet \leftarrow$ Submodular_Minimization($images$) ;   ▷ Select candidate set using submodular minimization
$verifiedSet \leftarrow$ manualVerification($candidateSet$) ;        ▷ Manually verify the candidate set

---

**Collection of Reference Images for Art Styles** We collect reference images for each artist (each artist is assumed to have a distinct art style) from Wikiart, the online encyclopedia for art works. Since the art works of each artist were meticulously collected and vetted by the artist community, we consider all the images collected from Wikiart as the reference art images for that artist.

# F   Implementation Details of MIMETIC$^2$ for Human Face Imitation

## F.1   Filtering of Training Images

Images whose captions mention the concept of interest often do not contain it (as shown with Mary-Lee Pfeiffer in Figure 3). As such, we filter images where the concept does not appear in the image, which we detect using a dedicated classifier. In what follows we describe the filtering mechanism.

**Collecting Reference Images:**

We collect reference images for each person using SerpAPI as described in Appendix E. These images are the gold standard images that we manually vet to ensure that they contain the target person of interest (see Appendix E for the details). We use the reference images to filter out the images in the pretraining dataset that are not of this person. Concretely, for each person we use a face embedding model (Deng et al., 2020) to measure the similarity between the faces in the reference images and the faces in the images from the

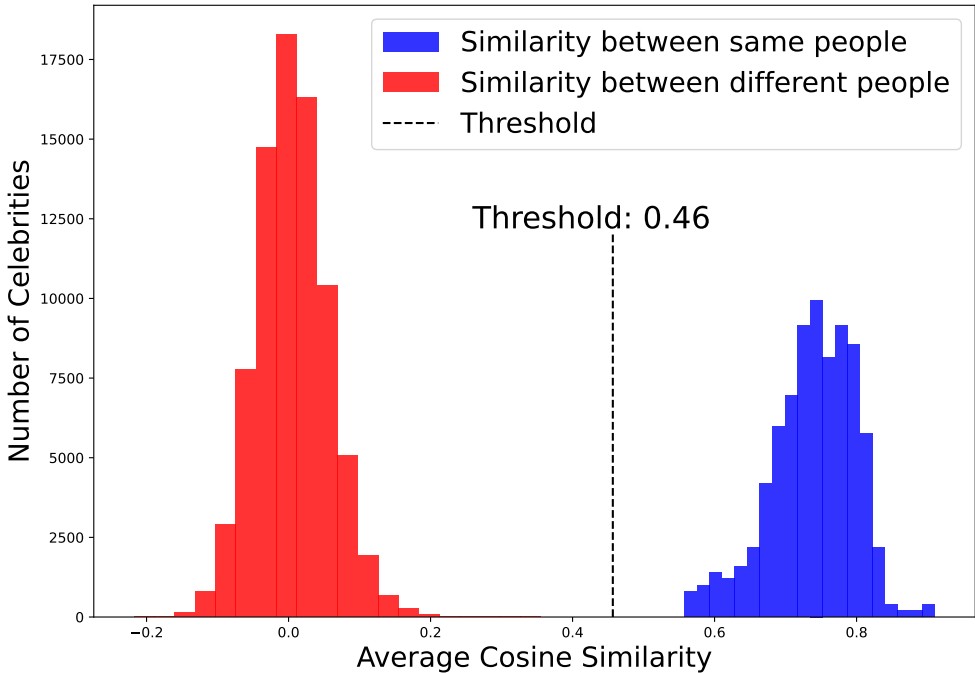

Figure 10: Average cosine similarity between the faces of the same people (blue colored) and of the faces of different people (red colored), measured across the reference images of the celebrities.

Table 8: Metrics for the classifier used to distinguish between the faces of individuals and the art style of artists.

| Domain | Accuracy (%) | Precision (%) | Recall (%) |
|---|---|---|---|
| Celebrity | 100 | 100 | 100 |
| Politicians | 100 | 100 | 100 |
| Classical Artists | 95.0 | 92.3 | 98.3 |
| Modern Artists | 94.4 | 92.9 | 96.1 |

pretraining datasets whose captions mention this person. If the similarity of a face in the pretraining images to any of the faces in the reference images is above a certain threshold, that face is considered to belong to the person of interest. We determine this threshold to distinguish faces of the same person from faces of different persons in the next paragraph. Note that this procedure already filter outs any image that does not contain a face, because the face embedding model would only embed an image if it detects a face in that image.

**Determining Filtering Threshold:** The next step is to determine the threshold for which we consider two faces to belong to the same person. For this purpose, we measure the similarity between pairs of faces of the same person and the similarity between pairs of faces of different persons Since the reference images for each person is manually vetted to be correct, we use these images for this procedure. We plot the histogram of the average similarity between the faces of the same person (blue colored) and the similarity between faces of different persons (red colored) in Figure 10. We see that the two histograms are well separated, with the lowest similarity value between the faces of the same person being 0.56 and the highest similarity value between the faces of different persons being 0.36. Therefore, any threshold value between 0.36 and 0.56 can separate two faces of the same person, from the faces of different people. In our experiments, we use the midpoint threshold of 0.46 (true positive rate (tpr) of 100%; false positive rate (fpr) of 0%) to filter any face in the pretraining images that do not belong to the person of interest (see Table 8 for the classifier metrics). The filtering process gives us both the image frequency a person in the pretraining data, and the pretraining images that we compare the faces in the generated images to measure the imitation score.

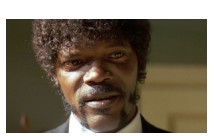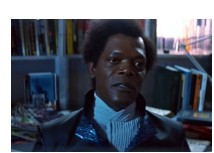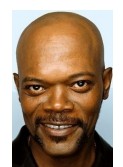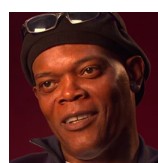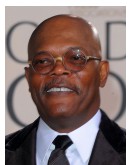

(a) Training images of *Samuel L. Jackson* that show significant variations in his face (age, hair, and beard).

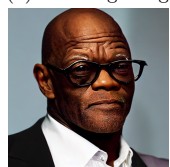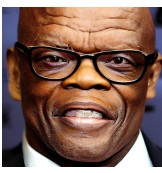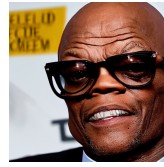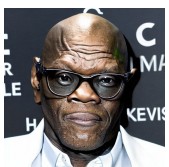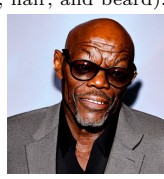

(b) Generated images of *Samuel L. Jackson* that show the model has captured a specific characteristic of his face (middle-aged, bald, with no or little beard).

Figure 11: Real and generated images of *Samuel L. Jackson*.

## F.2 Measurement of Imitation Score

To measure the imitation between the training and generated images of a person, we compute the cosine similarity between the face embeddings of the faces in their generated images and their filtered training images from the previous step. However, measuring the similarity using all the pretraining images can underestimate the actual imitation. This is because several individuals have significant variations in their faces in the pretraining images and the text-to-image model does not capture all these variations. For example, consider the pretraining images of *Samuel L. Jackson* in Figure 11a. These images have significant variations in beard, hair, and age. However, when the text-to-image model is prompted to generate images of *Samuel L. Jackson*, the generated images in Figure 11b only show a specific facial characteristic of him (middle-aged, bald, with no or little beard). Since MIMETIC$^2$'s goal is not to measure if a text-to-image model captures all the variations of a person, we want to reward the model even if it has only captured a particular characteristic (which it has in this case of *Samuel L. Jackson*). Therefore, instead of comparing the similarity of generated images to all the training images, we compare the similarity to only the ten training images that have the highest cosine similarity to the generated images on average.

## G Implementation Details of MIMETIC$^2$ for Art Style Imitation

### G.1 Filtering of Training Images

For art style imitation, we consider each artist to have a unique style. We collect the images from the pretraining dataset whose captions mention the name of the artist whose art style imitation we want to measure. Similar to the case of human face imitation, we want to filter out the pretraining images of an artist that in reality was not created by that artist, but their captions mention them. We implement the filtering process in two stages. In the first stage, we filter out non-art images in the pretraining dataset (note that the captions of these images still mention the artist, but the images themselves are not art works) and in the second stage we filter out art works of other artists (the captions of these images mention the artist of interest and the image itself is also an art work, but by a different artist). The implementation details for each stage is as follows:

**Filtering Non-Art Images:** To filter non-art images from the pretraining dataset, we use a classifier that separates art images from non-art images. Concretely, we embed the pretraining images using a CLIP ViT-H/14 (Ilharco et al., 2021) image encoder and measure the cosine similarity of the image embeddings and the text embeddings of the string '*an artwork*', embedded using the text encoder of the same model. Only when the similarity between the embeddings is higher than a threshold described below, we consider those pretraining images as an artwork. To determine this threshold, we choose a similarity score that separates art images from non-art images. We use the images from the Wikiarts dataset (Saleh & Elgammal, 2016) as the (positive) art images and MS COCO dataset images (Lin et al., 2014) as the (negative) non-art images. Note that MS COCO dataset was collected by photographing everyday objects that art was not part of, making it a valid set of negative examples of art.

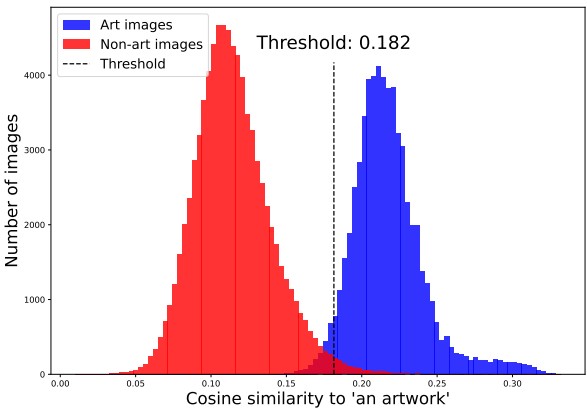 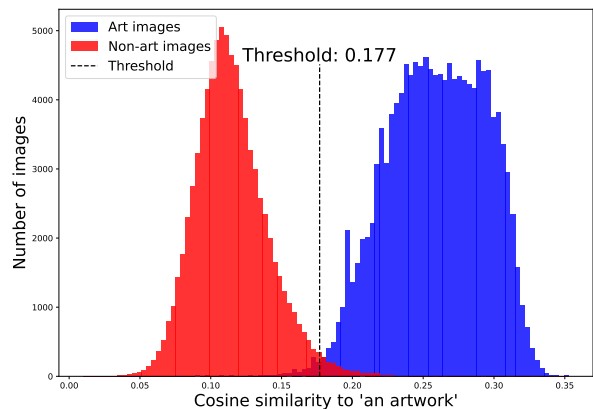

(a) Histogram of the cosine similarity of embeddings of art and non-art images to embeddings of 'an artwork' for classical artists.

(b) Histogram of the cosine similarity of embeddings of art and non-art images to embeddings of 'an artwork' for modern artists.

Figure 12: The first filtering step involves determining the threshold to distinguish between art and non-art images from the pretraining images, for which we compare the similarity of the image's embedding to the embedding of the text "an artwork".

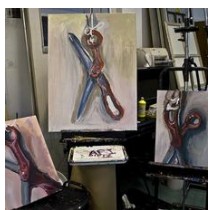 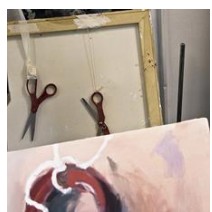 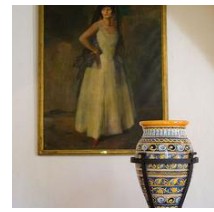 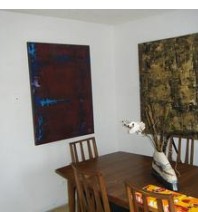 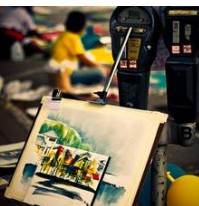

(a) Images from the MS COCO dataset that were classified as art by the threshold we choose. These images clearly have paintings in them and therefore are classified in that category. These images were selected in MS COCO for different categories like scissors, chair, parking meter, and vase.

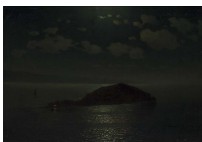 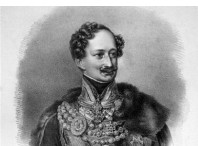 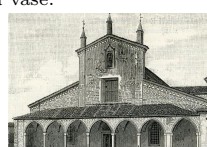 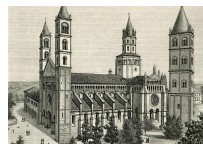 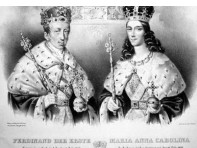

(b) Images from the Wikiarts dataset that were classified as non-art by the threshold we choose.

Figure 13: Images that are misclassified by our art vs. non-art threshold in Figure 12a.

We plot the histogram of cosine similarity of the embeddings of art and non-art images to the text embedding of '*an artwork*' (see Figures 12a and 12b. We observe that the art and non-art images both the artist groups are well separated (although not perfect, Figure 13 and Figure 14 shows examples of misclassified and correctly classfied images from both datasets). We choose the threshold that maximizes the F1 score of the separation (0.182 for the classical artists and 0.177 for the modern artists).

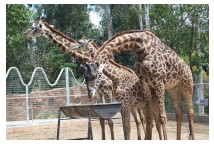 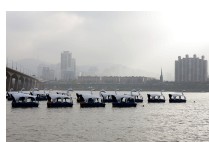 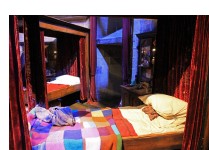 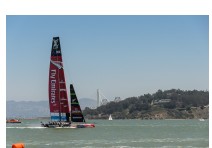 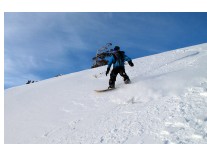

(a) Images from the MS COCO dataset that were correctly classified as non-art.

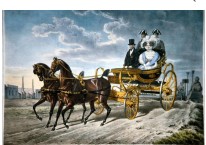 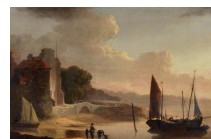 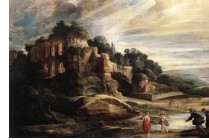 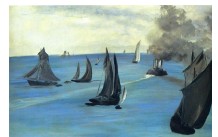 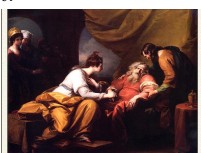

(b) Images from the Wikiarts dataset that were correctly classified as art.

Figure 14: Images that are correctly classified by our art vs. non-art threshold in Figure 12a.

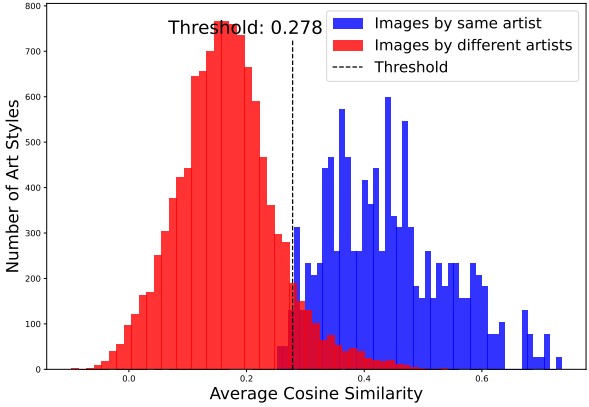 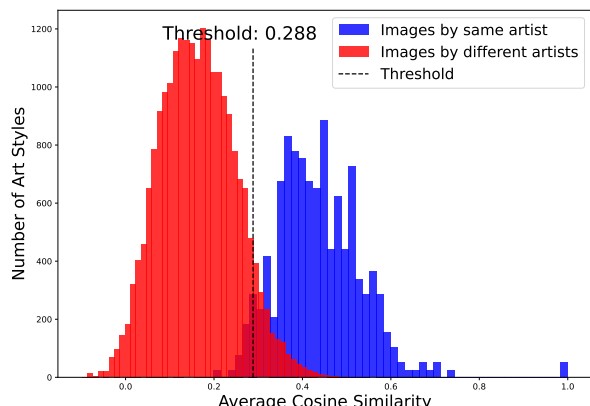

(a) Histogram of the average cosine similarity between embeddings of the images of the same artist (blue) and the art of different artists (red) for classical artists

(b) Histogram of the average cosine similarity between embeddings of the images of the same artist (blue) and the art of different artists (red) for modern artists

Figure 15: The second filtering step involves determining the if an art work whose caption mentions an artist actually belongs to that artist or not.

**Filtering Images of Other Art Styles:** Similar to the case of human faces, not all art images whose captions mention an artist were created by that artist. We want to filter out such images. For this purpose, we collect reference images for each artist (see Appendix E for details) and use them to classify the training images that belong to the artist of interest. Concretely, we measure the similarity between the pretraining images and the reference images of each artist, and only retain images whose similarity to the reference images is higher than a threshold.

To determine this threshold, we measure the similarity between pairs of art images of the same artists and pairs of art images from different artists. We embed the images using an art style embedding model (Somepalli et al., 2024) and plot the histogram of similarities between art images of the same artist (blue colored) and art images of different artists (red colored) in Figure 15a for classical artists and Figure 15b for modern artists. We see that the two histograms are well separated (although not perfect, Figure 16 shows paintings by two artists whose art style is very similar and cannot be distinguished by our threshold). We choose the threshold that maximizes the F1 score of the separation between these two groups (0.278 for classical artists and 0.288 for modern artists). See Table 8 for the classifier metrics. The retained images give us both the image counts of each artist and the training images that we compare to the generated images to measure the imitation score.

### G.2 Similarity Measurement

We embed all the generated images and the filtered pretraining images using the art style embedding model (Somepalli et al., 2024) and measure the cosine similarity between each pair of generated and pretraining images. Similar to the case of the human faces, we do not want to underestimate the art style similarity between the generated and training images by comparing the generated images to all the training images of this artist. Therefore, we measure the similarity of generated images to the ten training images that are on average the most similar to the generated images.

## H Evaluation of the Imitation Thresholds

Evaluating the accuracy of the imitation thresholds MIMETIC$^2$ finds is a very expensive task, it requires conducting the optimal experiment described in Section 3 (training $\mathcal{O}(\log m)$ models with different number of images of each concept). In order to make the evaluation tractable, we finetune a pretrained text-to-image model with concepts that the model has not seen during pretraining and find the number of images required for the model to be able to imitate them. We find that all the concepts require more than 250 images for imitation when the models are finetuned on them, which is very close to the imitation threshold MIMETIC$^2$

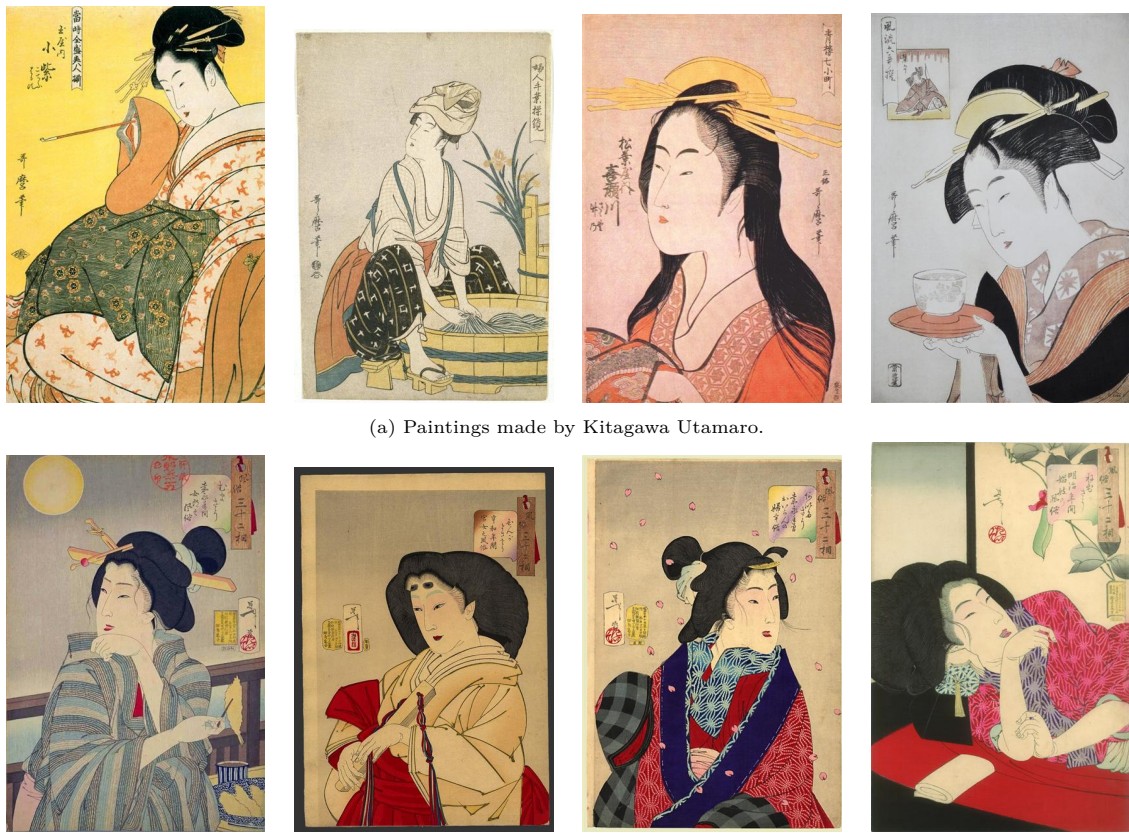

(a) Paintings made by Kitagawa Utamaro.

(b) Paintings made by Tsukioka Yoshitoshi

Figure 16: Paintings made by Kitagawa Utamaro and Tsukioka Yoshitoshi are very similar and our threshold is unable to distinguish between their styles.

finds for this pretrained model without finetuning its (234 images). Based on these results, we conclude that the imitation thresholds estimated by $\text{MIMETIC}^2$ are accurate.

We do this by searching politicians who became popular after the LAION-2B-en dataset was released (after 2022) (§C.2) and collecting their images from Google Images. We finetune SD1.4 on images of these politicians, starting with 50 images of a politician (much below the imitation threshold found by $\text{MIMETIC}^2$) and going up to 800 images (above the estimated imitation threshold). We finetune the model separately for each politician. To mimic the original training setup of the SD models (where images of a concept are naturally interspersed with other images in the dataset), we mix the politician images with other images taken randomly from the LAION-2B-en dataset such that for each finetuning experiment the total number of images added upto 10,000 (9,950 random images mixed with 50 images of the politician, 9,900 random images mixed with 100 images of the politician, and so on for each finetuning experiment). We finetune on this new dataset for a single epoch with a learning rate of $5e-5$.

We want to find the number of images required for imitation to occur in this setup. Imitation can be established either with a human study or an automatic metric that correlates with human perception of imitation. Since we already conducted a human study and found that our imitation threshold correlates with human perception, we argue that the automatic imitation score for concepts that are on the right hand side of the imitation threshold would also correlate with the human perception of imitation. $\text{MIMETIC}^2$ estimated that the imitation score for concepts on the right hand side of the imitation threshold for face imitation of politicians for SD1.4 is **0.26** (Figure 19). We deem the number of images required to reach this imitation score as the number of images required for imitation for this finetuning experiment.

We report the results in Table 9. It shows that the imitation score remains lower than **0.26** when less than 200 images are used to finetune the SD model and the score reaches 0.26 only when finetuned with 250 or more images of a politician.

Thus the number of images required to reach the imitation score of 0.26 in the finetuning experiment (250 images) is very close to the imitation threshold $\text{MIMETIC}^2$ finds for SD1.4 without requiring to finetune the model (234 images). Based on these results, we conclude that the imitation thresholds estimated by $\text{MIMETIC}^2$ are accurate.

Table 9: The imitation scores of the concepts as SD1.4 is finetuned on increasing number of images of a concept. We report imitation scores averaged over 10 such politicians. We note that the imitation scores have values 0.26 or greater when the model is finetuned on 250 or more images of a politician. Critically, this is the average imitation score for politicians on the right hand side of the imitation threshold in Figure 19 indicating that 250 images or more are required for imitation in this experiment.

| # Finetuning Images | 50 | 100 | 150 | 200 | 250 | 300 | 400 | 500 | 600 | 700 | 800 |
|---|---|---|---|---|---|---|---|---|---|---|---|
| Avg. Imitation Score | 0.07 | 0.16 | 0.19 | 0.21 | 0.26 | 0.27 | 0.29 | 0.31 | 0.32 | 0.36 | 0.34 |

# I  Imitation Thresholds of SD models in Series 1 and 2

Our experimental results in Section 6 found that for most domains the imitation thresholds for SD1.1 and SD1.5 are almost the same, while being higher for SD2.1. We hypothesized that the difference is due to their different text encoders. All models in SD1 series use the same text encoder from CLIP, whereas SD2.1 uses the text encoder from OpenCLIP. To test the validity of this hypothesis, we repeated the experiments for all models in SD1 series for politicians and computed their imitation thresholds. Table 10 shows the thresholds for the politicians. We find that the imitation thresholds for all the models in SD1 series is almost the same, and is lower than the threshold for SD2.1 model. This evidence supports our hypothesis of the difference in the text-encoders being the main reason for the difference in the imitation thresholds.

# J  Change Points

Table 11 we show all the change points that PELT found for each experiment (Table 3 reports the first change point as the imitation threshold).

Table 10: *Imitation Thresholds* for politicians for all models in SD1 series and SD2.1

| Pretraining Dataset | Model | Human Faces 👦: Politicians |
|---|---|---|
| LAION2B-en | SD1.1 | 234 |
| | SD1.2 | 252 |
| | SD1.3 | 234 |
| | SD1.4 | 234 |
| | SD1.5 | 234 |
| LAION-5B | SD2.1 | 369 |

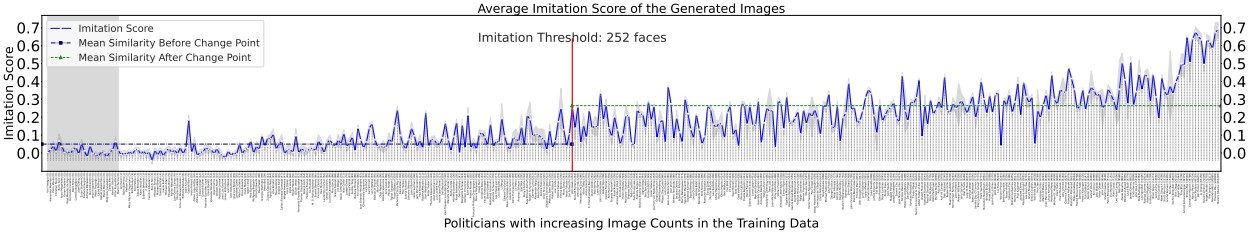

Figure 17: **Human Face Imitation (Politicians):** Similarity between the training and generated images for all politicians. The politicians with zero image counts are shaded with light gray. We show the mean and variance over the five generation prompts. The images were generated using **SD1.2**. The change point for human face imitation for politicians when generating images using SD1.1 is detected at **252 faces**.

# K    All Results: The *Imitation Threshold*

In this section, we estimate the imitation threshold for human face and art style imitation for three different text-to-image models. Figure 20, Figure 21, Figure 22, and Figure 23 show the image counts of celebrities on the x-axis (sorted in increasing order of image counts) and the imitation score of their generated images (averaged over the five image generation prompts) on the y-axis. The images were generated using LDM, SD1.1, SD1.5, and SD2.1 respectively. Similarity, Figure 24, Figure 25, Figure 26, and Figure 27 shows the image counts of the politicians and the imitation score of their generated images, for LDM, SD1.1, SD1.5, and SD2.1 respectively.

Figure 28, Figure 29, Figure 30, Figure 31 show the image counts of classical artists and the similarity between their training and generated images; and Figure 32, Figure 33, Figure 34, Figure 35 show the image counts of modern artists and the similarity between their training and generated images. The images were generated using LDM, SD1.1, SD1.5, and SD2.1 respectively.

**Imitation Threshold Estimation for Human Face Imitation:**   In Figure 21, we observe that the imitation scores for the individuals with small image counts is close to 0 (left side), and it increases as the number of their image counts increase towards the right. The highest similarity is 0.5 and it is for the individuals in the rightmost region of the plot. The solid line in the plot shows the mean similarity over the five image generation prompt with the shaded area showing the variance over them. We observe a low variance in the imitation score among the generation prompts. And we also observe that the variance does not depend on the image counts which indicates that the performance of the face recognition model does not depend on the popularity of the individual. The change detection algorithm finds the change point to be at **364** faces for human face imitation for celebrities, when using **SD1.1** for image generation. Figure 22 shows the similarity between the training and generated images when images are generated using **SD1.5**. Identically to SD1.1, the change is detected at **364** faces for face imitation when using SD1.5. We also performed ablation experiments with different face embeddings models and justify the choice of our model (see Appendix M). For all the plots, we also analyze the trend by using isotonic regression which learns non-decreasing linear regression weights that fits the data best.

**Imitation Threshold Estimation for Human Face Imitation (Politicians):**   Figure 25 shows the imitation scores for politicians which is very similar to the plot obtained for celebrities. We observe a low variance in the imitation score among the generation prompts. We also observe that the variance does not

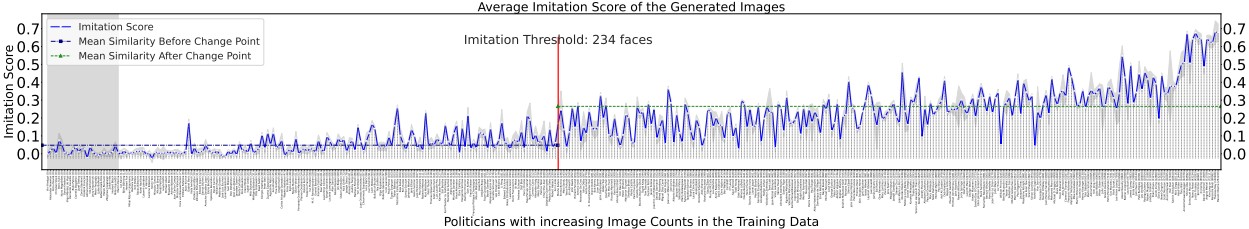

Figure 18: **Human Face Imitation (Politicians):** Similarity between the training and generated images for all politicians. The politicians with zero image counts are shaded with light gray. We show the mean and variance over the five generation prompts. The images were generated using **SD1.3**. The change point for human face imitation for politicians when generating images using SD1.1 is detected at **234 faces**.

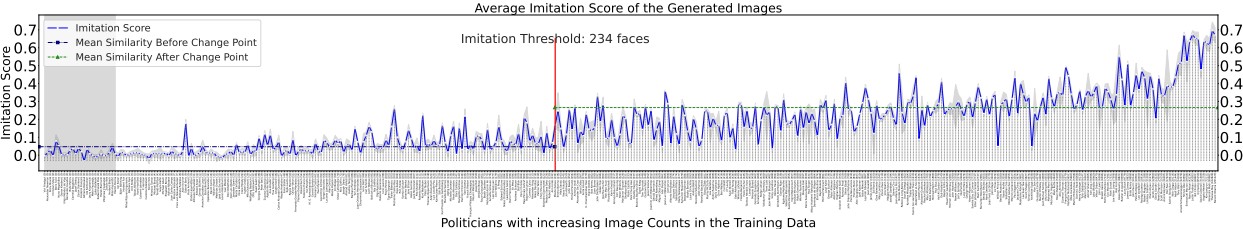

Figure 19: **Human Face Imitation (Politicians):** Similarity between the training and generated images for all politicians. The politicians with zero image counts are shaded with light gray. We show the mean and variance over the five generation prompts. The images were generated using **SD1.4**. The change point for human face imitation for politicians when generating images using SD1.1 is detected at **234 faces**.

depend on the image counts which indicates that the performance of the face recognition model does not depend on the popularity of the individual. The change detection algorithm finds the change point to be at **234** faces for human face imitation for politicians, when using **SD1.1** for image generation. Figure 26 shows the similarity between the training and generated images when images are generated using **SD1.5**. Similar to SD1.1, the change is detected at **234** faces.

**Imitation Threshold Estimation for Art Style Imitation:** In Figure 29, we observe that the imitation scores for artists with low image counts have a baseline value around 0.2 (left side), and it increases as the number of their image counts increase towards the right. The highest similarity is 0.76 and it is for the artists in the rightmost region of the plot. We also observe a low variance across the generation prompts, and the variance does not depend on the image frequency of the artist. The change detection algorithm finds the change point to be at **112** images for art style imitation of classical artists, when using **SD1.1** for image generation. Figure 30 shows the similarity between the training and generated images when images are generated using **SD1.5**. Similar to SD1.1, the change is detected at **112** faces for art style imitation when using SD1.5. These thresholds are slightly higher for style imitation of modern artists, 198 for both SD1.1 and SD1.5.

## L  Examples of Outliers

Figure 36 and Figure 37 show examples of outliers of the first kind, where aliases of a celebrity leads to under counting of their images in the pretraining data.

Figure 38 and Figure 39 show examples of outliers of the first kind for artists, where aliases of an artist leads to under counting of their art works in the pretraining data.

## M  Ablation Experiment with Different Face Embedding Models

In this section, we show the difference in the performance of several face embedding models and justify the choice of the final choice of our face embedding model. Face embedding models are evaluated using two main metrics: false-match rate (FMR) and true-match rate (TMR) (NIST, 2020). FMR measures how many times does a model says two people are the same when they are not and TMR measures how many times a model

Table 11: *Imitation Thresholds* for human face and art style imitation for the different text-to-image models and datasets we experiment with. This table shows all the change points that PELT found for each experiment (Table 3 reports the first change point as the imitation threshold).

| Pretraining Dataset | Model | Human Faces 🧑 | | Art Style 🖼️ | |
| --- | --- | --- | --- | --- | --- |
| | | Celebrities | Politicians | Classical Artists | Modern Artists |
| LAION2B-en | SD1.1 | 364 | 234 | 112, 391 | 198 |
| | SD1.5 | 364, 8571 | 234, 4688 | 112, 360 | 198, 4821 |
| LAION-5B | SD2.1 | 527, 9650 | 369, 8666 | 185, 848 | 241, 1132 |

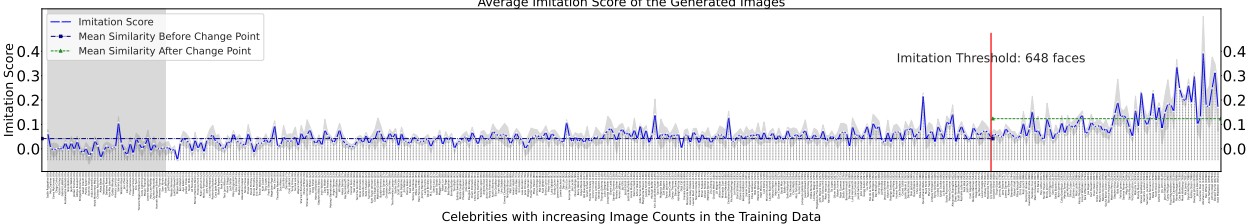

Figure 20: **Human Face Imitation (Celebrities):** Similarity between the training and generated images for all celebrities. The celebrities with zero image counts are shaded with light gray. We show the mean and variance over the five generation prompts. The images were generated using **LDM**. The change point for human face imitation for celebrities when generating images using LDM is detected at **648 faces**.

says two people are the same when they are the same. Ideally, a face embedding model should have low FMR and high TMR. An important variant of these metrics is the disparity of FMR and TMR of a model across different demographic groups. Ideally, a model should have low disparity in these metrics across different demographics. We also focus on the variance of these metrics across demographics in making the final choice.

We evaluate the FMR and TMR of eight different face embedding models (seven open-sourced and one proprietary). The open-source models were chosen based on their popularity on Github (Serengil & Ozpinar, 2020; Deng et al., 2022; 2020), and we also experiment with Amazon Rekognition, a proprietary model. For evaluating the disparity of these metrics across different demographic groups we grouped celebrities in six demographic groups primarily categorized according to skin color tone (black, brown, and white) and perceived gender (male and female; for simplicity). Each of the six groups had 10 celebrities (a total of 60), with no intersection between them. The categorization was done manually by looking at the reference images of the celebrities. For each celebrity, we collect 10 reference images from the internet by using the procedure described in Appendix E. We use these images to compare the FMR and TMR of the face recognition models, as these images are the gold standard images of a person.

**FMR Computation:** We compute the mean cosine similarity between the face embeddings of one individual and the faces of all other individuals in that group, and repeat the procedure for all individuals in a demographic group.

**TMR Computation:** We compute the mean cosine similarity between the embeddings of all the faces of an individuals and repeat the procedure for all the individuals in a demographic group.

Figure 40 and Figure 41 shows the FMR and TMR for six demographic groups for all the face embedding models. All the open-sourced models, except InsightFace, either have a high disparity in FMR values across the demographic groups (ArcFace, Facenet, Facenet512, DeepFace) or have very low TMR (GhostFaceNet_W1, GhostFaceNet_V1). We choose InsightFace for our experiments because of it has 1) a low overall FMR, 2) decent TMR, 3) a low disparity of FMR and TMR across the demographic groups, and 4) is open-sourced. Having a low disparity of the metrics across individuals of different demographic groups is crucial for an accurate estimation of the imitation threshold. The Amazon Rekognition model would also be a viable choice based on these metrics, however, it is not open-sourced and therefore expensive for our experiments.

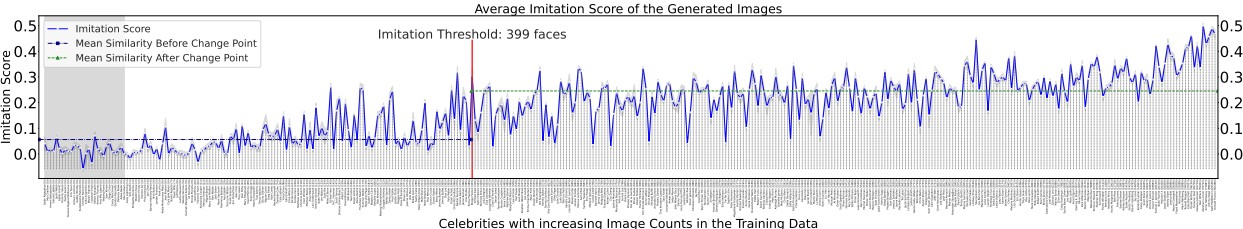

Figure 21: **Human Face Imitation (Celebrities):** Similarity between the training and generated images for all celebrities. The celebrities with zero image counts are shaded with light gray. We show the mean and variance over the five generation prompts. The images were generated using **SD1.1**. The change point for human face imitation for celebrities when generating images using SD1.1 is detected at **364 faces**.

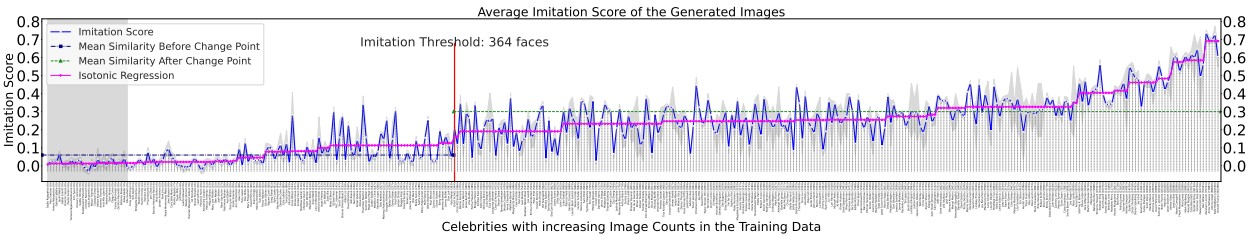

Figure 22: **Human Face Imitation (Celebrities):** similarity between the training and generated images for all celebrities. We show the mean and variance over the five generation prompts. The images were generated using **SD1.5**. The change point for human face imitation for celebrities when generating images using SD1.5 is detected at **364 faces**.

# N    Count Distribution and the List of Sampled Entities for Each Domain

## N.1    Celebrities

We collect celebrities from https://www.popsugar.com/Celebrities and https://celebanswers.com/celebrity-list/. The distribution of the caption counts of the sampled celebrities is displayed in Table 12. The sampled celebrities in the descending order of their number of caption counts are:

Donald Trump, Kate Middleton, Abraham Lincoln, Johnny Depp, Stephen King, Anne Hathaway, Ben Affleck, Ronald Reagan, Oprah Winfrey, Floyd Mayweather, Dwayne Johnson,
↪ Cameron Diaz, Cate Blanchett, Mark Wahlberg, Naomi Campbell, Nick Jonas, Jessica Biel, Kendrick Lamar, Malcolm X, Steven Spielberg, Bella Thorne, Bob Ross, Jay Leno,
↪ David Tennant, Samuel L. Jackson, Jason Statham, Mandy Moore, Victoria Justice, Scott Disick, Martin Scorsese, Ashley Olsen, Carey Mulligan, Greta Thunberg, Ashlee
↪ Simpson, Kacey Musgraves, Kurt Russell, Felicity Jones, Saoirse Ronan, Sarah Paulson, Matthew Perry, Forest Whitaker, Brendon Urie, Meg Ryan, Olivia Culpo, Joe Rogan,
↪ Sacha Baron Cohen, Terrence Howard, Natalie Dormer, Ansel Elgort, Nick Offerman, Clive Owen, Rose Leslie, Sterling K. Brown, Cuba Gooding Jr., Kevin James, Marisa Tomei,
↪ Troye Sivan, Zachary Levi, Gwendoline Christie, Hunter Hayes, Melanie Martinez, Joel McHale, Ross Lynch, Brody Jenner, Riley Keough, Robert Kraft, Ray Liotta, Eric Bana,
↪ Mark Consuelos, Chris Farley, James Garner, Lauren Daigle, Lily Donaldson, Penélope Cruz, Karen Elson, Joey Fatone, Leslie Odom Jr., Jay Baruchel, Selita Ebanks, Lana
↪ Condor, Mackenzie Foy, Doja Cat, Skai Jackson, Sofia Hellqvist, Bernard Arnault, Josh Peck, Lindsay Price, Phoebe Bridgers, Sarah Chalke, Alexander Skarsgård, Tai Lopez,
↪ Léa Seydoux, Cam Gigandet, David Dobrik, Jacob Elordi, Omar Epps, Marsai Martin, Alyson Stoner, Dree Hemingway, Gregg Sulkin, Mamie Gummer, Allison Holker, Chris Watts,
↪ Jacob Sartorius, Christine Quinn, Torrey Devitto, Alek Wek, Sandra Cisneros, Robert Irvine, Danielle Fishel, Normani Kordei, Sam Taylor Johnson, Jessica Seinfeld,
↪ Rachelle Lefevre, Joyner Lucas, Jimmy Buffet, John Wayne Gacy, Marvin Sapp, Ryan Guzman, Lindsay Ellingson, John Corbett, Michaela Coel, Hanne Gaby Odiele, Christiano
↪ Ronaldo, Scott Speedman, Addison Rae, Justice Smith, Stella Tennant, Lindsay Wagner, AJ Michalka, Charles Melton, Patricia Field, Dan Bilzerian, Annie Murphy, Michiel
↪ Huisman, Sara Foster, Diego Boneta, Danny Thomas, Oliver Hudson, Lauren Bushnell, Chris Klein, Rodrigo Santoro, Luke Hemsworth, Rhea Perlman, Michael Peña, Jodie
↪ Turner-Smith, Trevor Jackson, Jenna Marbles, Bob Morley, Zak Bagans, Liza Koshy, Steve Lacy, Nico Tortorella, Emma Corrin, Lo Bosworth, Quvenzhané Wallis, Martin Starr,
↪ David Muir, Beanie Feldstein, Lori Harvey, Eddie McGuire, Todd Chrisley, Dan Crenshaw, Amanda Gorman, Crystal Renn, Mark Richt, Magdalena Frackowiak, Danielle Jonas, Liu
↪ Yifei, Sasha Pivovarova, Ashleigh Murray, Peter Hermann, Daria Strokous, Eddie Hall, Hunter Parrish, Matt McGorry, Diane Guerrero, Simu Liu, Brady Quinn, Jill Wagner,
↪ Richard Rawlings, Sophia Lillis, Genesis Rodriguez, Diane Ladd, Frankie Grande, Olivia Rodrigo, Anwar Hadid, Hannah Bronfman, Deana Carter, Tao Okamoto, Fei Fei Sun,
↪ Taylor Tomasi Hill, Jared Followill, Margherita Missoni, Elisa Sednaoui, Thomas Doherty, Bill Skarsgård, Indya Moore, Ziyi Zhang, Cacee Cobb, Jay Ellis, Arthur Blank,
↪ Chris McCandless, Paz de la Huerta, Jacquelyn Jablonski, Michael Buffer, Annie LeBlanc, Kieran Culkin, Lacey Evans, Rachel Antonoff, Presley Gerber, Lauren Bush Lauren,
↪ Peter Firth, Tina Knowles-Lawson, Sunisa Lee, Douglas Brinkley, Hero Fiennes-Tiffin, Erin Foster, Justina Machado, Mariacarla Boscono, Summer Walker, Emma Chamberlain,
↪ Lew Alcindor, Jenna Ortega, Phoebe Dynevor,
    Kim Zolciak-Biermann, Allison Stokke, Malgosia Bela, Isabel Toledo, Sydney Sweeney, Mat Fraser, Hunter McGrady, Ethan Suplee, Tammy Hembrow, Ivan Moody, Danneel Harris,
↪ Marcus Lemonis, Hunter Schafer, Luka Sabbat, Sam Elliot, Kendra Spears, Stephen tWitch Boss, Joe Lacob, Tommy Dorfman, Emma Barton, Elliot Page, Sha'Carri Richardson,
↪ Barry Weiss, Julie Chrisley, Devon Sawa, Miles Heizer, Julia Stegner, Austin Abrams, Jacquetta Wheeler, Melanie Iglesias, Anna Cleveland, Eiza González, Grant Achatz,
↪ Matt Stonie, Connor Cruise, Nicholas Braun, Dan Lok, Charli D'Amelio, Jeremy Bamber, Jim Walton, Matthew Bomer, Nicola Coughlan, Una Stubbs, Andrew East, Miles O'Brien,
↪ Mary Fitzgerald, Taylor Mills, Portia Freeman, Kate Chastain, David Brinkley, Bregje Heinen, DJ Kool Herc, Barbie Ferreira, Paul Mescal, Forrest Fenn, Jamie Bochert, Yung
↪ Gravy, Daisy Edgar-Jones, Dixie D'Amelio, Jordan Chiles, Bob Keeshan, Alexandra Cooper, Kyla Weber, Chase Stokes, Belle Delphine, Joanna Hillman, Olivia O'Brien, Jillie
↪ Mack, Maggie Rizer, Sasha Calle, Tony Lopez, Danny Koker, Irwin Winkler, M.C. Hammer, Zack Bia, Alexa Demie, Bailey Sarian, Yael Cohen, Angie Varona, Trevor Wallace,
↪ Madelyn Cline, Fred Stoller, Frank Sheeran, Albert Lin, Sessilee Lopez, Zaya Wade, Maitreyi Ramakrishnan, Madison Bailey, Will Reeve, Nick Bolton, Rege-Jean Page, Matthew
↪ Garber, Yamiche Alcindor, Isaak Presley, Thandiwe Newton, Nicole Fosse, Shenae Grimes-Beech, Alex Choi, Scott Yancey, Ciara Wilson, Lexi Underwood, Manny Khoshbin, Ella
↪ Emhoff, Cole LaBrant, Wayne Carini, Greg Fishel, Ryan Upchurch, Marcus Freeman, Danielle Cohn, Sue Aikens, Kyle Cooke, David Portnoy, Avani Gregg, Dan Peña, Quinton
↪ Reynolds, Eric Porterfield, Ayo Edebiri, Tara Lynn Wilson, Florence Hunt, Nicola Porcella, Pashmina Roshan, Josh Seiter, Ben Mallah, Miguel Bezos, Lukita Maxwell, Ali
↪ Skovbye, Jordan Firstman, Jeff Molina, Mary Lee Pfeiffer, Cody Lightning, Leah Jeffries, Elle Graham, Hannah Margaret Selleck, Woody Norman, Tom Blyth, Banks Repeta,
↪ Wisdom Kaye, Kris Tyson, Joey Klaasen, Tioreore Ngatai-Melbourne, Jani Zhao, Cara Jade Myers, Keyla Monterroso Mejia, Samara Joy, Mason Thames, Park Ji-hu, Boman
↪ Martinez-Reid, Priya Kansara, Yasmin Finney, Bridgette Doremus, Aria Mia Loberti, Isabel Gravitt, Gabriel LaBelle, Delaney Rowe, Armen Nahapetian, Aditya Kusupati, Vedang
↪ Raina, Arsema Thomas, Adwa Bader, Amaury Lorenzo, Corey Mylchreest, Sam Nivola, Gabby Windey, Cwaayal Singh, Jaylin Webb, Kudakwashe Rutendo, Chintan Rachchh, Sajith
↪ Rajapaksa, Diego Calva, Pardis Saremi, Dominic Sessa, India Amarteifio, Mia Challiner, Aryan Simhadri

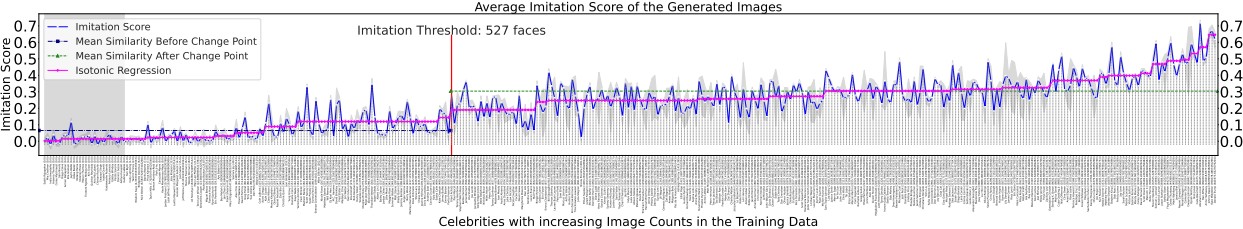

Figure 23: **Human Face Imitation (Celebrities):** similarity between the training and generated images for all celebrities. We show the mean and variance over the five generation prompts. The images were generated using **SD2.1**. The change point for human face imitation for celebrities when generating images using SD2.1 is detected at **527 faces**.

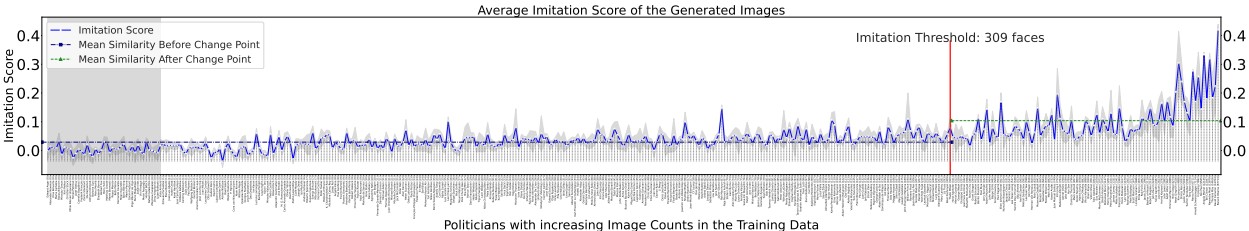

Figure 24: **Human Face Imitation (Politicians):** Similarity between the training and generated images for all politicians. The politicians with zero image counts are shaded with light gray. We show the mean and variance over the five generation prompts. The images were generated using **LDM**. The change point for human face imitation for politicians when generating images using LDM is detected at **309 faces**.

## N.2    Politicians

We collect politicians from Wikipedia ([Wikipedia, 2024](#)). The distribution of the caption counts of the sampled politicians is given in Table [12](#). The sampled politicians in the descending order of their number of caption counts are:

Barack Obama, John Lewis, Theresa May, Narendra Modi, Kim Jong-un, David Cameron, Angela Merkel, Bill Clinton, Xi Jinping, Justin Trudeau, Emmanuel Macron, Nancy Pelosi,
↪ Arnold Schwarzenegger, Ron Paul, Shinzo Abe, Adolf Hitler, John Paul II, Tony Blair, Sachin Tendulkar, Nick Clegg, Newt Gingrich, Scott Morrison, Arvind Kejriwal, Ilham
↪ Aliyev, Jacob Zuma, Bashar al-Assad, Laura Bush, Sonia Gandhi, Kim Jong-il, Robert Mugabe, James Comey, Rodrigo Duterte, Pete Buttigieg, Lindsey Graham, Hosni Mubarak,
↪ Enda Kenny, Alexei Navalny, Rob Ford, Leo Varadkar, Evo Morales, Lee Hsien Loong, Henry Kissinger, Petro Poroshenko, Joko Widodo, Clarence Thomas, Rishi Sunak, Mohamed
↪ Morsi, Ashraf Ghani, Martin McGuinness, Viktor Orban, Uhuru Kenyatta, Mike Huckabee, Sheikh Hasina, Martin Schulz, Giuseppe Conte, John Howard, Benito Mussolini, Tulsi
↪ Gabbard, Dominic Raab, Michael D. Higgins, François Hollande, Yasser Arafat, Mark Rutte, Mahathir Mohamad, Juan Manuel Santos, Abiy Ahmed, William Prince, Lee Kuan Yew,
↪ Mikhail Gorbachev, Hun Sen, Jacques Chirac, Martin O'Malley, Benazir Bhutto, Yoshihide Suga, John Major, Muammar Gaddafi, Jerry Springer, Sandra Day O'Connor, Madeleine
↪ Albright, Thomas Mann, Paul Kagame, Simon Coveney, Grant Shapps, Sebastian Coe, Merrick Garland, Jean-Yves Le Drian, Nursultan Nazarbayev, Horst Seehofer, Liz Truss,
↪ Rowan Williams, Ellen Johnson Sirleaf, George Weah, Mark Sanford, Yoweri Museveni, Luigi Di Maio, Ben Wallace, Herman Van Rompuy, Daniel Ortega, Olaf Scholz, Beppe
↪ Grillo, Alassane Ouattara, Nicolás Maduro, Tamim bin Hamad Al Thani, Mary McAleese, Asif Ali Zardari, Joseph Goebbels, Nikol Pashinyan, Deb Haaland, Paul Biya, Abdel
↪ Fattah el-Sisi, Thabo Mbeki, Kyriakos Mitsotakis, Joseph Muscat, Micheál Martin, Rebecca Long-Bailey, Paschal Donohoe, Todd Young, Jean-Marie Le Pen, Nick Griffin, Zoran
↪ Zaev, Pierre Nkurunziza, Abhisit Vejjajiva, Maggie Hassan, Steven Chu, Juan Guaidó, Edi Rama, Mary Landrieu, Jyrki Katainen, Jens Spahn, John Dramani Mahama, Gina
↪ Raimondo, Alec Douglas-Home, Viktor Orbán, Anita Anand, Isaias Afwerki, James Cleverly, Ibrahim Mohamed Solih, Leymah Gbowee, Václav Havel, John Rawls, Jack McConnell,
↪ Romano Prodi, Eoghan Murphy, Vicky Leandros, Norodom Sihamoni, Nayib Bukele, Shirin Ebadi, Jusuf Kalla, George Eustice, Joachim von Ribbentrop, Peter Altmaier, Akbar
↪ Hashemi Rafsanjani, Paul Singer, Christian Stock, Moussa Faki, Dominique de Villepin, Michael Fabricant, Kim Dae-jung, Eamon Ryan, Shavkat Mirziyoyev, Denis
↪ Sassou-Nguesso, Werner Faymann, Kamla Persad-Bissessar, Ingrid Betancourt, Volodymyr Zelenskyy, Park Chung Hee, Elvira Nabiullina, Roselyne Bachelot, Heinz Fischer,
↪ Hideki Tojo, Anatoly Karpov, Marcelo Ebrard, Slavoj Žižek, Trent Lott, Alfred Rosenberg, Gabi Ashkenazi, Valentina Matviyenko, Kgalema Motlanthe, Pedro Castillo, Winona
↪ LaDuke, Peter Bell, Boyko Borisov, Carl Bildt, Almazbek Atambayev, Andry Rajoelina, Carl Schmitt, Ralph Gonsalves, Liam Byrne, Alok Sharma, Jean-Michel Blanquer, Robert
↪ Schuman, Shinzō Abe, Doris Leuthard, Jacques Delors, Floella Benjamin, Sauli Niinistö, Annalena Baerbock, Toomas Hendrik Ilves, Alejandro Giammattei, Bob Kerrey, Lionel
↪ Jospin, Murray McCully, Stefan Löfven, Javier Solana, Salva Kiir Mayardit, Cecil Williams, Shahbaz Bhatti, Marianne Thyssen, Marty Natalegawa, Roh Moo-hyun, John
↪ Diefenbaker, Antonio Inoki, Iván Duque, CY Leung, Tom Tancredo, Sigrid Kaag, Jim Bolger, Lou Barletta, Li Peng, Laura Chinchilla, Gennady Zyuganov, Chen Shui-bian,
Sebastián Piñera, Gustavo Petro, Miguel Díaz-Canel, Alberto Fernández, Gerald Darmanin, Boutros Boutros-Ghali, Joschka Fischer, Maia Sandu, Ricardo Martinelli, Andrej Babiš,
↪ Dan Jarvis, Nikos Dendias, Chris Hipkins, Tawakkol Karman, Booth Gardner, Karin Kneissl, Mobutu Sese Seko, Alexander Haig, Alexander De Croo, Ahmed Aboul Gheit, Yasuo
↪ Fukuda, Jean-Luc Mélenchon, Jane Ellison, Diane Dodds, Helen Whately, Idriss Déby, Patrice Talon, Carmen Calvo, Dario Franceschini, Emma Bonino, Richard Ferrand, Andreas
↪ Scheuer, Moshe Katsav, K. Chandrashekar Rao, P. Harrison, Robert Habeck, Ann Linde, Jon Ashworth, Edward Scicluna, Stef Blok, Lawrence Gonzi, William Roper, Josep Rull,
↪ Sam Kutesa, Raja Pervaiz Ashraf, David Cairns, Ilir Meta, Perry Christie, Rinat Akhmetov, Ahmet Davutoğlu, Franck Riester, Nikos Christodoulides, Damien O'Connor, Sali
↪ Berisha, Umberto Bossi, Lee Cheuk-yan, Alpha Condé, Alexander Newman, Annette Schavan, Yuri Andropov, Peter Tauber, Faure Gnassingbé, Bolkiah of Brunei, Karl-Theodor zu
↪ Guttenberg, Michael Brand, Helen Suzman, Ron Huldai, Mohamed Azmin Ali, François-Philippe Champagne, Agostinho Neto, Marielle de Sarnez, Kurt Waldheim, Mounir Mahjoubi,
↪ Juan Orlando Hernández, Angela Kane, Lech Wałęsa, Luis Lacalle Pou, Barbara Pompili, Margaritis Schinas, Tigran Sargsyan, Wolfgang Bosbach, Raed Saleh, Johanna Wanka,
↪ Michelle Donelan, Roberto Speranza, Traian Băsescu, Iurie Leancă, Dara Calleary, Ilona Staller, Micheline Calmy-Rey, Thomas Oppermann, Karine Jean-Pierre, Luciana
↪ Lamorgese, Azali Assoumani, Michael Adam, Paulo Portas, Svenja Schulze, Pita Sharples, Choummaly Sayasone, Federico Franco, Félix Tshisekedi, Roberta Metsola, Nia
↪ Griffith, Paul Myners, Ahmad Vahidi, Kaja Kallas, Hua Guofeng, Olga Rypakova, Otto Grotewohl, Audrey Tang, Oskar Lafontaine, Ivica Dačić, Isa Mustafa, Xiomara Castro, M.
↪ G. Ramachandran, Fernando Grande-Marlaska, Wopke Hoekstra, Tomáš Petříček, Egils Levits, Roland Koch, Joseph Deiss, Laurentino Cortizo, Alan García, Nikola Poposki,
↪ Evarist Bartolo, Reyes Maroto, Zuzana Čaputová, Sergei Stanishev, Plamen Oresharski, Ana Brnabić, Carlos Alvarado Quesada, Marek Biernacki, Olivier Véran, Vjekoslav
↪ Bevanda, Clare Moody, Matthias Groote, Giorgos Stathakis, Marta Cartabia, Elena Bonetti, Dina Boluarte, Milo Đukanović, Levan Kobiashvili, Isabel Celaá, Jarosław Gowin,
↪ José Luis Escrivá, Cora van Nieuwenhuizen, Ivan Mikloš, Arancha González Laya, Viola Amherd, Gernot Blümel, José Luis Ábalos, Deo Debattista, Alain Krivine, Zlatko
↪ Lagumdžija, Edward Argar, Adrian Năstase, Zdravko Počivalšek, Miroslav Kalousek, Gabriel Boric, Juan Carlos Campo, Karel Havlíček, Kiril Petkov, Elżbieta Rafalska, Tobias
↪ Billström, Miroslav Toman, Mihai Răzvan Ungureanu, Ivaylo Kalfin, Élisabeth Borne, Herbert Fux, Petru Movilă, Koichi Tani, Caroline Edelstam, Barbara Gysi, Ľubomír
↪ Jahnátek, Nuno Magalhães, Martin Pecina, Goran Knežević, Björn Böhning, Iñigo Méndez de Vigo, Božo Petrov, Ian Karan, Hernando Cevallos, Milan Kujundžić, Adriana Dăneasă,
↪ Ida Karkiainen, Zoran Stanković, Boris Tučić, Jerzy Kropiwnicki, Rafael Catalá Polo, Ljube Boškoski, Camelia Bogdănici, Józef Oleksy, Frederik François, Zbigniew Ć
↪ wiąkalski, Herbert Bösch, Metin Feyzioğlu, Zoltán Illés, Vivi Friedgut

## N.3    Classical Artists

We collected classical artists from the [https://www.wikiart.org](https://www.wikiart.org), a website that collects various arts from different artists and categorizes them into pre-defined art style categories. For classical artists, we collected

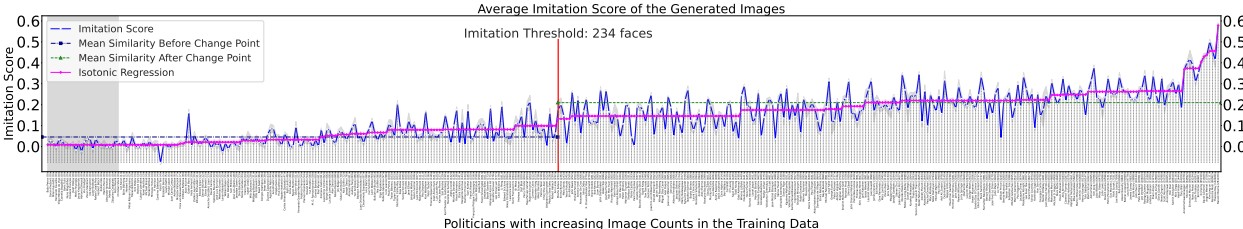

Figure 25: **Human Face Imitation (Politicians):** Similarity between the training and generated images for all politicians. The politicians with zero image counts are shaded with light gray. We show the mean and variance over the five generation prompts. The images were generated using **SD1.1**. The change point for human face imitation for politicians when generating images using SD1.1 is detected at **234 faces**.

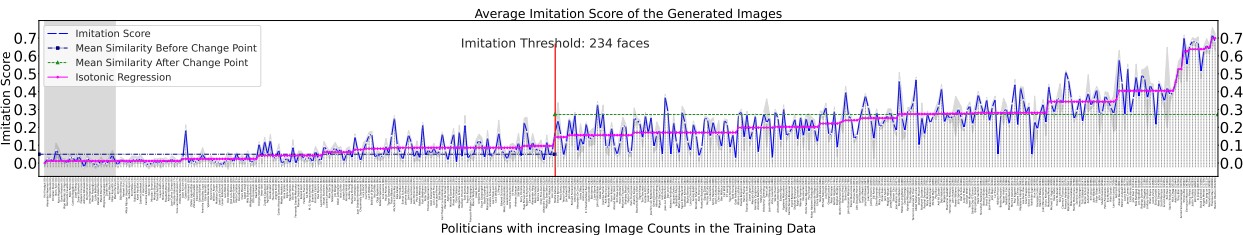

Figure 26: **Human Face Imitation (Politicians):** similarity between the training and generated images for all politicians. We show the mean and variance over the five generation prompts. The images were generated using **SD1.5**. The change point for human face imitation for politicians when generating images using SD1.5 is detected at **234 faces**.

the artist names from the art styles: *Romanticism, Impressionism, Realism, Baroque, Neoclassicism, Rococo, Academic Art, Symbolism, Cubism, Naturalism.* The distribution of the caption counts of the sampled artists is given in Table 12. The sampled artists in the descending order of their number of caption counts are:

> Claude Monet, Rembrandt, Gustav Klimt, Edgar Degas, Caravaggio, William Blake, John James Audubon, Le Corbusier, Canaletto, Peter Paul Rubens, John Singer Sargent, Edouard Manet, John William Waterhouse, Alfred Sisley, Childe Hassam, Berthe Morisot, Victor Hugo, William-Adolphe Bouguereau, Gustave Courbet, Albert Bierstadt, Mary Cassatt, John Constable, Gustave Dore, Gustave Caillebotte, Henry Moore, Thomas Hardy, Johannes Vermeer, Jacques-Louis David, Odilon Redon, Thomas Cole, Thomas Moran, James Tissot, William Hogarth, David Roberts, Thomas Gainsborough, Anthony Van Dyck, William Merritt Chase, Caspar David Friedrich, Sir Lawrence Alma-Tadema, George Stubbs, Georges Braque, Auguste Rodin, Joshua Reynolds, John Atkinson Grimshaw, David James, James Ward, David Johnson, Frederic Edwin Church, Jean-Leon Gerome, Eugene Delacroix, Martin Johnson Heade, Edward Burne-Jones, John William Godward, James Webb, Gustave Moreau, James Charles, Francois Boucher, Francisco Goya, John Everett Millais, Thomas Lawrence, John Ruskin, John Russell, David Davies, Dante Gabriel Rossetti, George Henry, John Martin, Frans Hals, Guido Reni, George Catlin, Claude Lorrain, Anders Zorn, Jessie Willcox Smith, Giovanni Battista Tiepolo, Howard Pyle, Archibald Thorburn, Thomas Eakins, Giovanni Boldini, Armand Guillaumin, Ivan Aivazovsky, John Trumbull, Joseph Wright, Benjamin West, John Collier, Henri Fantin-Latour, Jan Steen, Eugene Boudin, James Mcneill Whistler, Ilya Repin, William Bradford, Julia Margaret Cameron, Annibale Carracci, Antoine Watteau, Marianne North, David Cox, Jacob Jordaens, Frederick Morgan, Ivan Shishkin, George Morland, Ford Madox Brown, Frans Snyders, John Jackson, Aelbert Cuyp, Charles Willson Peale, Jacob Van Ruisdael, Joseph Ducreux, Horace Vernet, Pieter De Hooch, Arthur Hughes, Antonio Canova, Charles Le Brun, Francesco Hayez, Thomas Sully, Isaac Levitan, Robert Spencer, Karl Bodmer, Alexandre Cabanel, N.C. Wyeth, Anna Ancher, Carl Spitzweg, David Wilkie, Paul Delaroche, Charles-Francois Daubigny, George Frederick Watts, Guy Rose, Carel Fabritius, Alfred Stevens, Peder Severin Kroyer, Taras Shevchenko, Pietro Longhi, Joaquín Sorolla, Theodore Chasseriau, John Riley, Theodore Rousseau, Edmund Charles Tarbell, Giovanni Domenico Tiepolo, Edward Ladell, Pompeo Batoni, Richard Parkes Bonington, Boris Kustodiev, Andreas Achenbach, Charles Conder, Viktor Vasnetsov, Antoine Blanchard, William Henry Hunt, Emile Claus, Julian Alden Weir, Mikhail Vrubel, Richard Dadd, Vasily Vereshchagin, John Hoppner, Richard Lindner, Aristide Maillol, Joan Blaeu, William Williams, Adriaen Brouwer, Constant Troyon, Fernand Khnopff, Edwin Austin Abbey, Gino Severini, Pietro Da Cortona, Adriaen Van De Velde, Vasily Perov, David Bomberg, Konstantin Korovin, Christoffer Wilhelm Eckersberg, Jean Metzinger,
>
> Konstantin Makovsky, Mihaly Munkacsy, Albert Pinkham Ryder, Francesco Solimena, Franz Richard Unterberger, Roger De La Fresnaye, William Shayer, Paul Bril, Cornelis Springer, Jacques Lipchitz, Agostino Carracci, Adam Elsheimer, Giuseppe De Nittis, Jules Joseph Lefebvre, Albert Gleizes, Willard Metcalf, Vasily Surikov, Giovanni Fattori, Lyubov Popova, Kuzma Petrov-Vodkin, Johan Christian Dahl, Jehan Georges Vibert, Mikhail Nesterov, Antoine Pesne, Konstantin Yuon, Hugo Simberg, Gerard Terborch, Alexander Ivanov, Eustache Le Sueur, Giuseppe Maria Crespi, Ferdinand Bol, Max Slevogt, Philip Wilson Steer, Osias Beert, Vasily Polenov, John Crome, Edward Poynter, Nicolae Grigorescu, Louis Marcoussis, Marcus Stone, Jacques Stella, Edmonia Lewis, Antonietta Brandeis, Konstantin Somov, Hendrick Terbrugghen, Cornelis De Vos, Charles Spencelayh, Ivan Kramskoy, Rudolf Von Alt, Philipp Otto Runge, Carolus-Duran, Ralph Earl, Eugene Carriere, Julius Leblanc Stewart, Ippolito Caffi, John Peter Russell, Jean Baptiste Vanmour, Antonio Mancini, Petrus Van Schendel, Benjamin Brown, Max Klinger, Ludwig Knaus, Maurice Braun, Vincenzo Camuccini, Jean-Étienne Liotard, Henry Tonks, Jacek Malczewski, Rubens Santoro, Pieter Codde, Jean-Paul Laurens, Louise Moillon, Jan Siberechts, David Morier, John Pettie, Felicien Rops, Leon Bonnat, Theodule Ribot, William Logsdail, Richard Jack, Homer Watson, François Gérard, Robert Julian Onderdonk, Lionel Noel Royer, Charles Gleyre, Anne Brigman, Thomas Jones Barker, Antonio Ciseri, Joseph Anton Koch, Anton Melbye, Nicolas Tournier, Peter Nicolai Arbo, Lev Lagorio, Matthias Stom, Jean-Baptiste Van Loo, Konstantinos Volanakis, Cornelis Vreedenburgh, Henryk Siemiradzki, Frederick George Cotman, Eva Gonzales, Jan Cossiers, Julio Gonzalez, Vladimir Makovsky, Fyodor Bronnikov, Paul Peel, Thomas Pollock Anshutz, Raden Saleh, Robert Lewis Reid, Joseph-Marie Vien, Arno Breker, Frederick William Burton, Ion Andreescu, Jankel Adler, William Leighton Leitch, Esaias Van De Velde, Dirck Van Baburen, Jacob Van Strij, Franz Stuck, Giovanni Battista Gaulli, Hans Gude, Harriet Backer, Christina Robertson, Thomas Francis Dicksee, Fyodor Vasilyev, Claudio Coello, Gustave Boulanger, Nikolaos Gyzis, George Ault, Francisco Herrera, John Lewis Krimmel, Marie Bashkirtseff, Sebastien Bourdon, Jacob Ochtervelt, Christen Kobke, Paul Gavarni, Edouard Debat-Ponsan, Gregoire Boonzaier, Dmitry Levitzky, André Gill, Julian Ashton, Telemaco Signorini, Orest Kiprensky, Fyodor Rokotov, Nicolas Toussaint Charlet, Pieter Saenredam, Henri-Pierre Picou, Johan Hendrik Weissenbruch, Émile Friant, Herbert Gustave Schmalz, Jean-Baptiste Pigalle, T. C. Steele, Arturo Michelena, Wilhelm Von Kaulbach, Algernon Talmage, Giovanni Costa, Paul Leroy, Ivan Vladimirov, Hermann Hendrich, Magnus Enckell, Pavel Fedotov, Ethel Carrick, Vincenzo Irolli, Leopold Survage, Lady Frieda Harris, Joseph Duplessis, Charles Maurin, Philip De Laszlo, Peter Fendi, Marie-Guillemine Benoist, Antonio Paoletti, Christian Wilhelm Allers, Tranquillo Cremona, Antonio Donghi, Penry Williams, Miklos Barabas, Alfred Concanen, Albert Maignan, Dobri Dobrev, Bertalan Szekely, Mariano Benlliure, Anton Azbe, Johannes Moreelse, Nicolae Vermont, Heinrich Bürkel, Jane Sutherland, Laslett John Pott, Petro Kholodny, Alexey Zubov, Eliseu Visconti, Pieter Wenning, Henri Le Fauconnier, Paul Ackerman, Armand Henrion, Ipolit Strambu, George Hemming Mason, Vilhelms Purvitis, Mykola Yaroshenko, Pavel Svinyin, Gustav Adolf Mossa, Fyodor Solntsev, Pedro Américo, Klavdy Lebedev, Ivan Milev, Albert Benois, Alexandre Antigna, George Demetrescu Mirea, Giulia Lama, Aurelio Tiratelli, Konstantin Vasilyev, Domenico Fiasella, David Kakabadze, Cornelis Van Noorde, Panos Terlemezian, Alexei Korzukhin, Maurice Poirson, Joaquin Agrasot, Toby Edward Rosenthal, Heinrich Papin, Vasile Popescu, Jérôme-Martin Langlois, Karl Edvard Diriks, Adam Van Der Meulen, Vsevolod Maksymovych, Leo Leuppi, Matej Sternen, Filippo Cifariello, Apollinary Goravsky, Pasquale Celommi, Giuseppe Barberis, Francesco Didioni, Octav Angheluta, Vytautas Kairiukstis, Gevorg Bashindzhagian, Serhij Schyschko, Noè Bordignon, Armando Montaner Valdueza, Alexander Clarot, Rosario Weiss Zorrilla, Vasyl Hryhorovych Krychevsky, Fernand Combes, Francesco Ribalta, Jean Alexandru Steriadi, Johann Baptist Clarot, Corneliu Michailescu, Nzante Spee

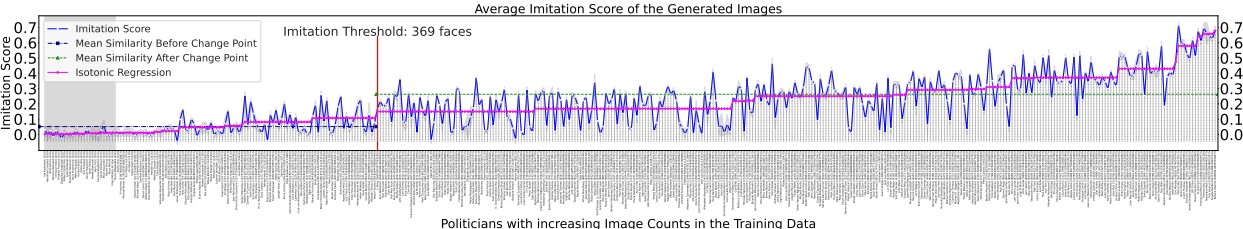

Figure 27: **Human Face Imitation (Politicians):** similarity between the training and generated images for all politicians. We show the mean and variance over the five generation prompts. The images were generated using **SD2.1**. The change point for human face imitation for celebrities when generating images using SD2.1 is detected at **369 faces**.

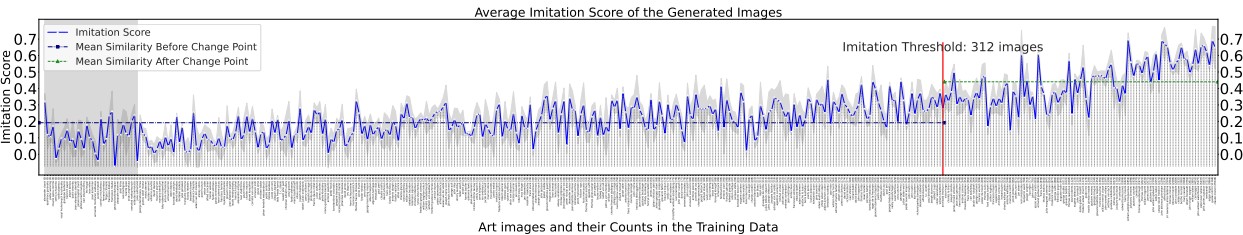

Figure 28: **Art Style Imitation (Classical Artists):** similarity between the training and generated images for **classical** art styles. We show the mean and variance over the five generation prompts. The images were generated using **LDM**. The change point for art style imitation when generating images using LDM is detected at **312 images**.

## N.4 Modern Artists

We also collected modern artists from the https://www.wikiart.org. For modern artists, we collected the artist names from the art styles: *Expressionism, Surrealism, Abstract Expressionism, Pop Art, Art Informel, Post-Painterly Abstraction, Neo-Expressionism, Post-Minimalism, Neo-Impressionism, Neo-Romanticism, Post-Impressionism.* The distribution of the caption counts of the sampled artists is given in Table 12. The sampled artists in the descending order of their number of caption counts are:

Vincent Van Gogh, David Bowie, Andy Warhol, Pablo Picasso, Frida Kahlo, Keith Haring, Salvador Dali, Paul Gauguin, Camille Pissarro, Paul Cezanne, Henri Matisse, Paul
↪ Klee, Francis Bacon, Edvard Munch, Amedeo Modigliani, Egon Schiele, Jean-Michel Basquiat, David Lynch, Wassily Kandinsky, Peter Max, Roy Lichtenstein, Paul Reed, Franz
↪ Marc, David Smith, Mark Rothko, Georges Seurat, Leroy Neiman, Joan Miro, Jackson Pollock, August Macke, Man Ray, Piet Mondrian, Cy Twombly, Henri De Toulouse-Lautrec,
↪ Graham Bell, Paul Signac, Robert Indiana, Yayoi Kusama, Rene Magritte, Jasper Johns, Walter Crane, Robert Morris, Emily Carr, Lucian Freud, Ernst Ludwig Kirchner, Tom
↪ Thomson, Anish Kapoor, Alex Katz, Pierre Bonnard, John Cage, Jim Dine, Ellsworth Kelly, Peter Blake, William Scott, Erin Hanson, Marcel Duchamp, Frank Stella, Robert
↪ Motherwell, Max Weber, Louise Nevelson, Peter Phillips, Willem De Kooning, Corneille, Wayne Thiebaud, Joan Mitchell, Jean Cocteau, Raoul Dufy, Antony Gormley, Max Ernst,
↪ Alberto Giacometti, Vanessa Bell, Richard Diebenkorn, James Rosenquist, Edouard Vuillard, Richard Hamilton, M.C. Escher, Sam Francis, Sean Scully, Anselm Kiefer, Edward
↪ Weston, Karel Appel, Philip Guston, Julian Schnabel, Ray Parker, James Ensor, Balthus, George Segal, Francis Picabia, Emil Nolde, Georges Rouault, Alice Neel, Helen
↪ Frankenthaler, Claes Oldenburg, Theo Van Rysselberghe, Maya Lin, Maurice Utrillo, Eric Fischl, H.R. Giger, Maurice Denis, Friedensreich Hundertwasser, Will Barnet, Paula
↪ Modersohn-Becker, Suzanne Valadon, Bruce Nauman, El Anatsui, Lee Krasner, Joseph Cornell, Patrick Heron, James Brooks, Paula Rego, Paul Jenkins, Jules Pascin, Lyonel
↪ Feininger, Edward Ruscha, Norman Lewis, Barnett Newman, David Park, Marie Laurencin, Rufino Tamayo, Chris Ofili, Lynd Ward, Jean-Paul Riopelle, Roger Fry, Remedios Varo,
↪ Maxime Maufra, Paul Serusier, Jacob Epstein, Richard Deacon, Walter Sickert, Mark Tobey, Jan Toorop, Jacek Yerka, Red Grooms, Ad Reinhardt, Eva Hesse, Oskar Kokoschka,
↪ Michael Sowa, Jean David, Sam Gilliam, Phyllida Barlow, Howard Finster, Augustus John, Elaine De Kooning, Beauford Delaney, David Hammons, Erich Heckel, Amrita Sher-Gil,
↪ Arthur Lismer, Mona Hatoum, Etel Adnan, Brion Gysin, John Chamberlain, Corita Kent, Allen Jones, Asger Jorn, Martin Kippenberger, George Tooker, Desmond Morris, Wolf
↪ Kahn, Jay Defeo, Irma Stern, Walasse Ting, Emile Bernard, Kathe Kollwitz, Frank Bowling, John Heartfield, Auguste Herbin, Frances Hodgkins, Meret Oppenheim, Andre Masson,
↪ Karl Schmidt-Rottluff, Hans Bellmer, Marino Marini, Morris Louis, Pyotr Konchalovsky, Richard Artschwager, Louis Cane, Betty Parsons, Max Pechstein, Richard
↪ Pousette-Dart, Georges Lemmen, Cuno Amiet, Louis Valtat, Kit Williams, Grace Cossington Smith, John Hoyland, Dennis Oppenheim, Lynda Benglis, James Lee Byars, Boris
↪ Grigoriev, Lili Elbe, Victor Brauner, Adrian Ghenie, Gillian Ayres, Ossip Zadkine, Alice Bailly, Felix Gonzalez-Torres, Johannes Itten, Charles Long, John Marin, Winifred
↪ Nicholson, Alfred Kubin, Charles Angrand, Zinaida Serebriakova, John Duncan Fergusson, Norman Bluhm,
        Harald Sohlberg, Zdzislaw Beksinski, Barkley L. Hendricks, Bruno Schulz, Toyen, Pierre Alechinsky, Hippolyte Petitjean, Nicolas De Staël, Rainer Fetting, Hiro Yamagata,
↪ Lorser Feitelson, Taro Yamamoto, Kazuo Shiraga, Alberto Burri, Anne Truitt, Jozsef Rippl-Ronai, Ronald Davis, Tsuguharu Foujita, Wols, Keith Sonnier, Henry Van De Velde,
↪ Chang Dai-Chien, Stanley Whitney, Ann Hamilton, John Brack, Jules-Alexandre Grun, Billy Apple, Eileen Agar, Benny Andrews, Moise Kisling, Edward Wadsworth, Paul Thek,
↪ Audrey Flack, Allan D'Arcangelo, Rik Wouters, Charles Cottet, Gene Davis, Prudence Heward, Alexander Liberman, David Batchelor, Tadanori Yokoo, Frederick Sommer, Hedda
↪ Sterne, Othon Friesz, Roderic O'Conor, Santiago Rusinol, Richard Gerstl, Marianne Von Werefkin, Octavio Ocampo, Kay Sage, Jessica Stockholder, Gabriele Munter, Jean
↪ Benoit, May Wilson, Jean Paul Lemieux, Jack Tworkov, Abraham Manievich, Perle Fine, Renato Guttuso, Al Held, Martial Raysse, Le Pho, Charles Reiffel, Bernard Cohen,
↪ Rosalyn Drexler, Ilya Mashkov, Jack Youngerman, Ernst Wilhelm Nay, Adja Yunkers, Leon Spilliaert, Valerio Adami, Karl Benjamin, Luigi Serafini, Leon Polk Smith, Oswaldo
↪ Guayasamin, Blinky Palermo, Ferdinand Du Puigaudeau, Esteban Vicente, Matsutani, Oscar Dominguez, Yasuo Kuniyoshi, Sergei Parajanov, Joy Hester, Forrest Bess, Taro
↪ Okamoto, Maurice Tabard, Yiannis Moralis, Igor Grabar, Alex Hay, Albert Irvin, Amadeo De Souza-Cardoso, Robert Swain, Bradley Walker Tomlin, Kishio Suga, Carlos Almaraz,
↪ Manuel Alvarez Bravo, Dan Christensen, Cyril Power, Marcel Barbeau, Jeremy Moon, Jorge Castillo, Josef Capek, Maggie Laubser, John Altoon, Albert Dubois-Pillet, William
↪ Baziotes, Joseph Marioni, Michael Hafftka, Raoul Ubac, Tony Feher, Walter Battiss, Friedel Dzubas, Varlin, Alfred Manessier, Ron Gorchov, Tony Scherman, Alina
↪ Szapocznikow, Nikolaos Lytras, Carl Holsøe, Constantin Brâncuşi, Walter Osborne, Max Kurzweil, Jose Guerrero, Leon Underwood, Istvan Nagy, Albert Bloch, Ward Jackson,
↪ Piero Dorazio, Giorgio Griffa, Lourdes Castro, Lita Albuquerque, Thomas Downing, Pierre Tal-Coat, Mario Prassinos, Panayiotis Tetsis, Robert Goodnough, Paul Feeley,
↪ Michel Majerus, Marc Vaux, Konstantinos Maleas, Vladimir Dimitrov, Meijer De Haan, Guido Molinari, Arthur Beecher Carles, Bertalan Por, Christo Coetzee, Jammie Holmes,
↪ Lasar Segall, Enrico Donati, Jerzy Nowosielski, Gianfranco Baruchello, Luis Feito, Burhan Dogancay, Iosif Iser, Charles Gibbons, Thalia Flora-Karavia, Aldo Mondino,
↪ Pierre Daura, Josef Sima, Nikola Tanev, Konrad Klapheck, Theophrastos Triantafyllidis, Edvard Weie, Gerard Fromanger, Matthias Laurenz Gräff, Victor Servranckx, Istvan
↪ Farkas, Ramon Oviedo, Manabu Mabe, Grégoire Michonze, Stanisław Ignacy Witkiewicz, Abidin Dino, Esteban Frances, Alberto Sughi, Olga Albizu, Behjat Sadr, Jose De
↪ Guimaraes, Robert Nickle, Dale Hickey, Inigo Manglano-Ovalle, Antonio Carneiro, Horia Damian, Jacqueline Hick, Kuno Gonschior, Huguette Arthur Bertrand, Ethel Léontine
↪ Gabain, Helen Dahm, Ion Nicodim, Lucy Ivanova, Gil Teixeira Lopes, Michel Carrade, Florin Maxa, Jean-Paul Jerome, Vangel Naumovski, Graca Morais, Antonio Areal, Petros
↪ Malayan, Rodolfo Arico, Stefan Sevastre, Johannes Sveinsson Kjarval, Ilka Gedo, Lucia Demetriade Balacescu, Natalia Dumitresco, Rene Bertholo, Vasile Kazar, Petre
↪ Abrudan, Aurel Cojan, Tia Peltz, Alvaro Lapa

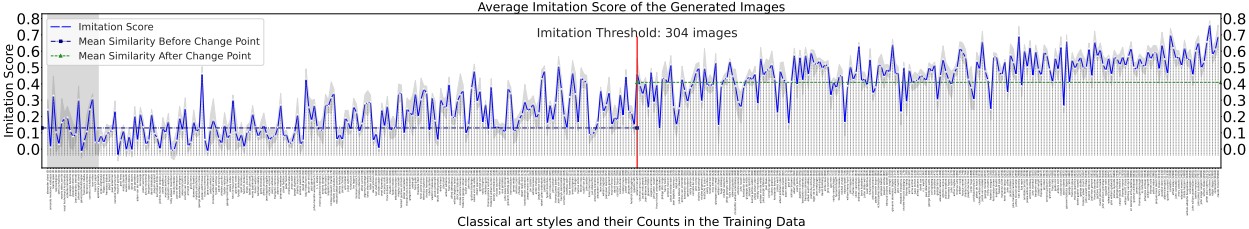

Figure 29: **Art Style Imitation (Classical Artists):** similarity between the training and generated images for **classical** art styles. We show the mean and variance over the five generation prompts. The images were generated using **SD1.1**. The change point for art style imitation when generating images using SD1.1 is detected at **112 images**.

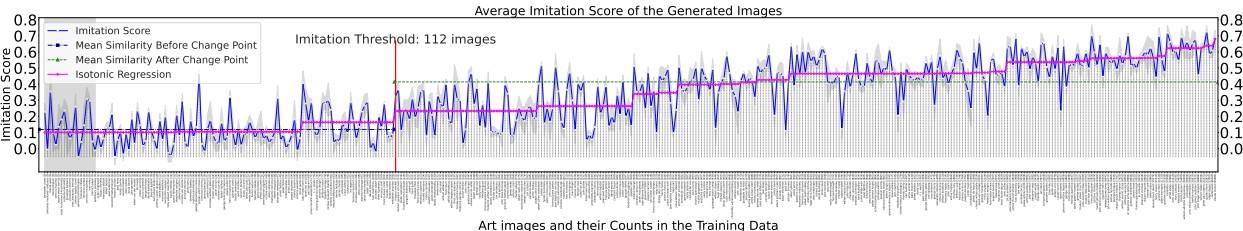

Figure 30: **Art Style Imitation (Classical Artists):** similarity between the training and generated images for **classical** art styles. We show the mean and variance over the five generation prompts. The images were generated using **SD1.5**. The change point for art style imitation when generating images using SD1.5 is detected at **112 images**.

## O  Compute Used

**Estimated Computational Cost of the Optimal Experiment**  The computational cost to train the popular text-to-image model Stable Diffusion was \$600K (Bastian, 2022). For the optimal experiment, we would need to train $\mathcal{O}(\log m)$ models, where $m$ is the total number of available images of a concept $Z^j$. So if a concept has 100,000 images, then we need to train $\log_2(100,000) = 16$ models to optimally estimate the imitation threshold, which would cost about \$10M.

**MIMETIC²'s Computational Cost**  We use 8 L40 GPUs to generate images for the all text-to-image models in our work. Overall, we use them for 16 hours per prompt, per dataset, per model to generate images. We downloaded the images on the same machine using 40 CPU cores, a process that took about 8 hours per dataset. For generating the image embeddings, we use the same 8 L40 GPUs, a process that took about 16 hours per dataset. The computation of imitation score and plotting are done on single CPU core on the same machine, a process that takes less than 30 minutes per dataset.

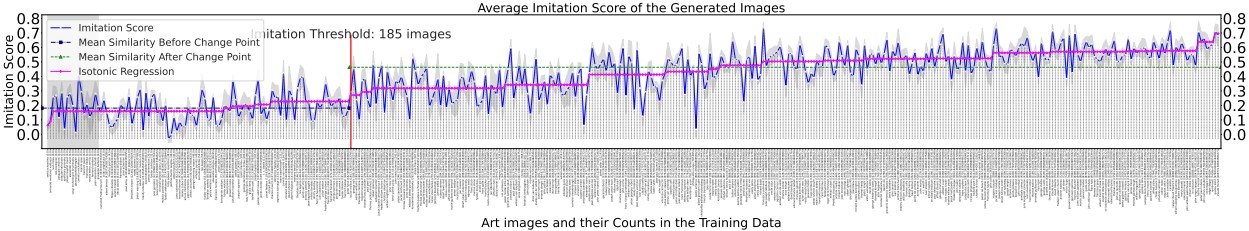

Figure 31: **Art Style Imitation (Classical Artists):** similarity between the training and generated images for **classical** art styles. We show the mean and variance over the five generation prompts. The images were generated using **SD2.1**. The change point for art style imitation when generating images using SD2.1 is detected at **185 images**.

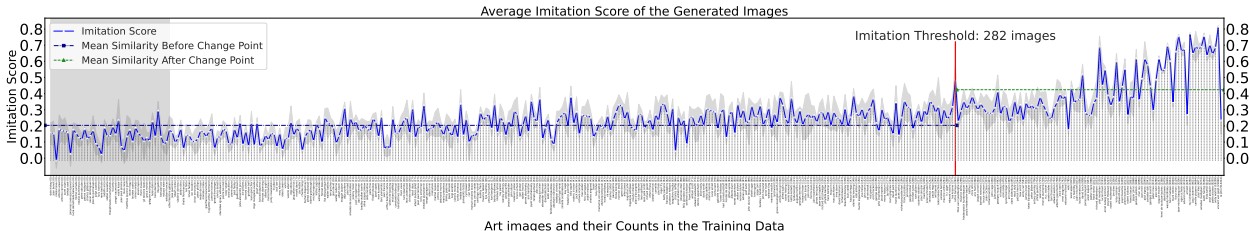

Figure 32: **Art Style Imitation (Modern Artists):** similarity between the training and generated images for **modern** art styles. We show the mean and variance over the five generation prompts. The images were generated using **LDM**. The change point for art style imitation when generating images using SD0 is detected at **282 images**.

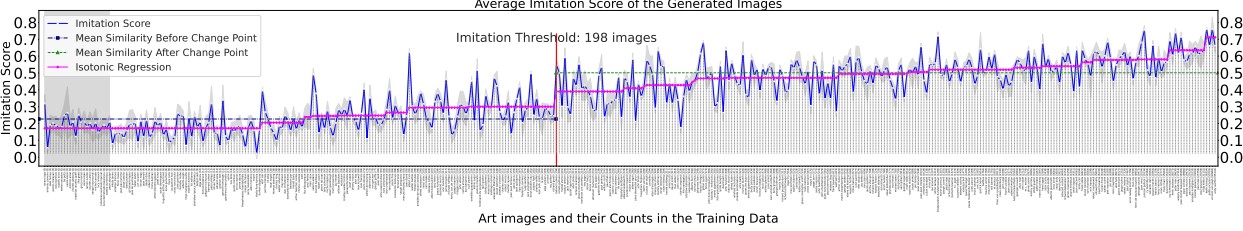

Figure 33: **Art Style Imitation (Modern Artists):** similarity between the training and generated images for **modern** art styles. We show the mean and variance over the five generation prompts. The images were generated using **SD1.1**. The change point for art style imitation when generating images using SD1.1 is detected at **198 images**.

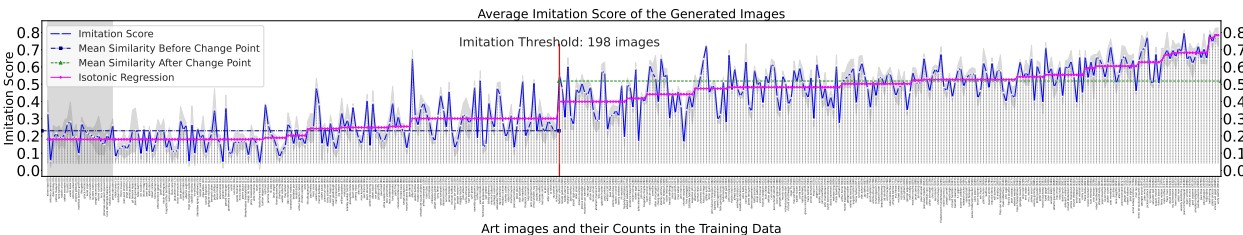

Figure 34: **Art Style Imitation (Modern Artists):** similarity between the training and generated images for **modern** art styles. We show the mean and variance over the five generation prompts. The images were generated using **SD1.5**. The change point for art style imitation when generating images using SD1.5 is detected at **198 images**.

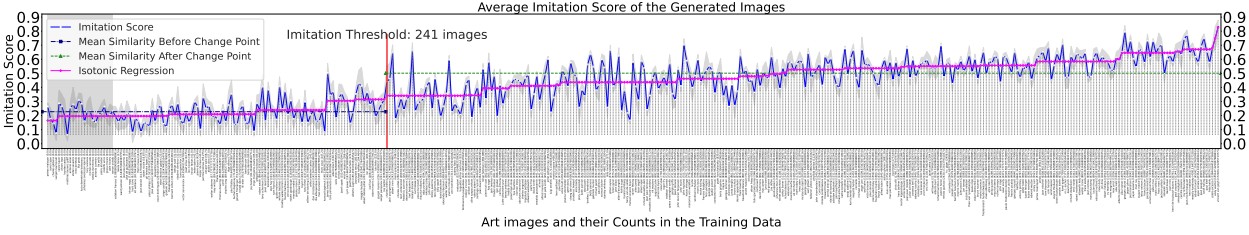

Figure 35: **Art Style Imitation (Modern Artists):** similarity between the training and generated images for **modern** art styles. We show the mean and variance over the five generation prompts. The images were generated using **SD2.1**. The change point for art style imitation when generating images using SD2.1 is detected at **241 images**.

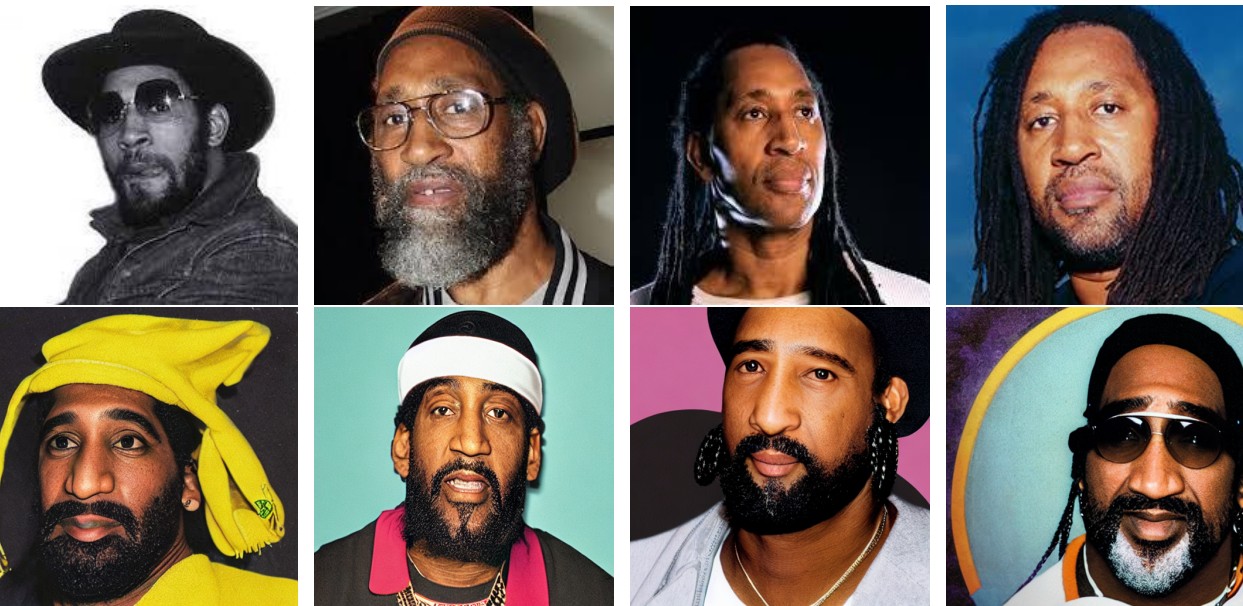

Figure 36: **Outlier Category 1: DJ Kool Herc.** *Clive Campbell* is aliased as *DJ Kool Herc*, which leads to lower counts of his images in the dataset since MIMETIC[2] only collects images whose caption mentions *DJ Kool Herc*.

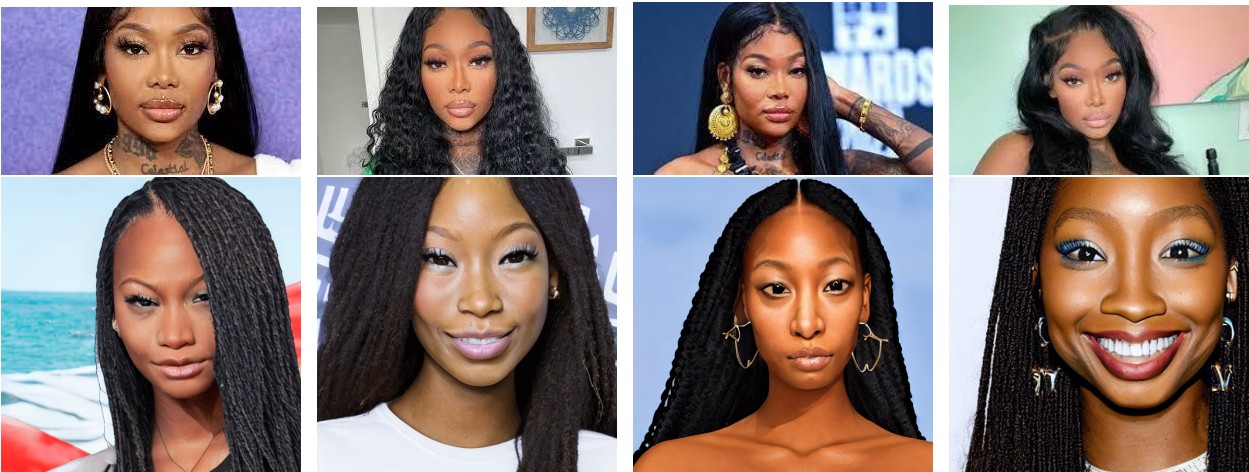

Figure 37: **Outlier Category 1: Summer Walker.** *Summer Marjani Walker* is aliased as *Summer Walker*, which leads to lower counts of her images in the dataset since MIMETIC[2] only collects images whose caption mentions *Summer Walker*.

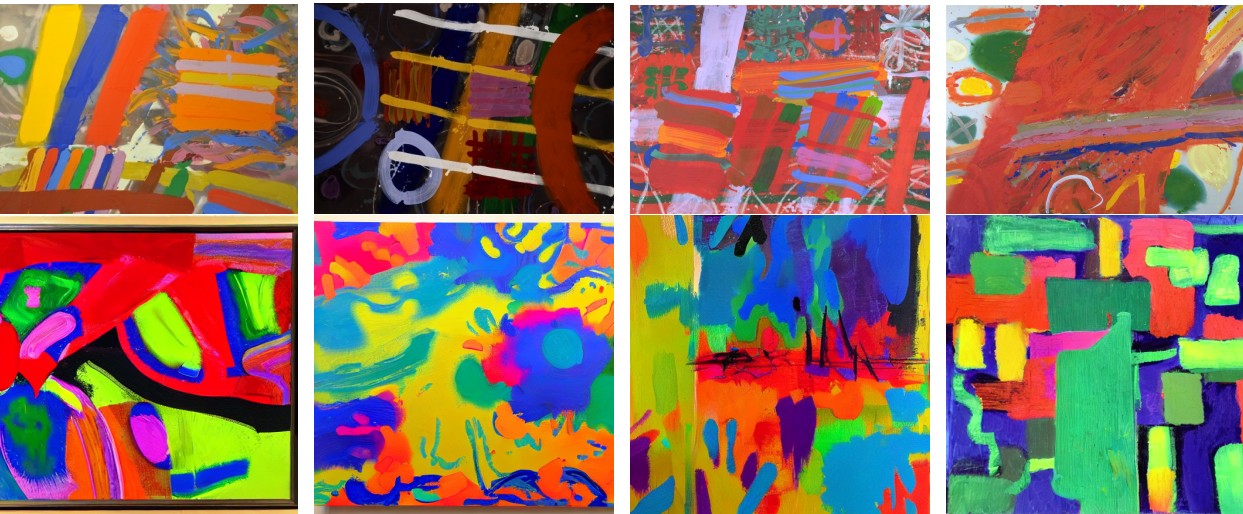

Figure 38: **Outlier Category 1: Albert Irwin.** *Albert Henry Thomas Irvin* is aliased as *Albert Irwin*, which leads to lower counts of his art images in the dataset since MIMETIC$^2$ only collects images whose caption mentions *Albert Irwin*.

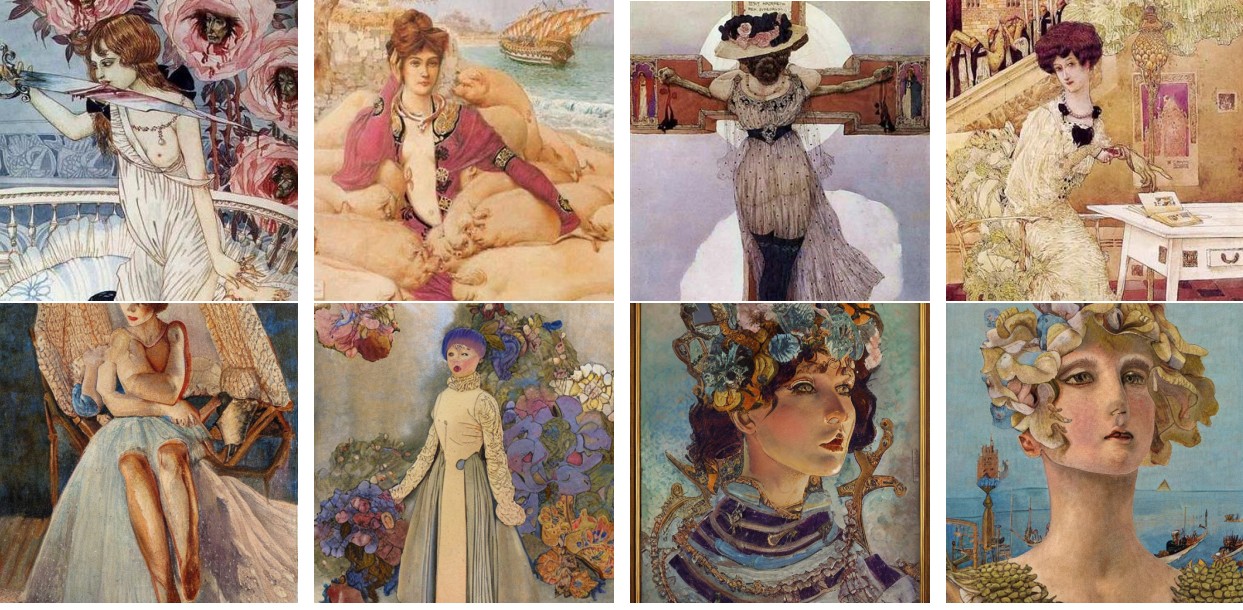

Figure 39: **Outlier Category 1: Gustav Adolf Mossa.** *Gustav Adolf Mossa* is aliased just as *Mossa*, which leads to lower counts of his art images in the dataset since MIMETIC$^2$ only collects images whose caption mentions *Gustav Adolf Mossa*.

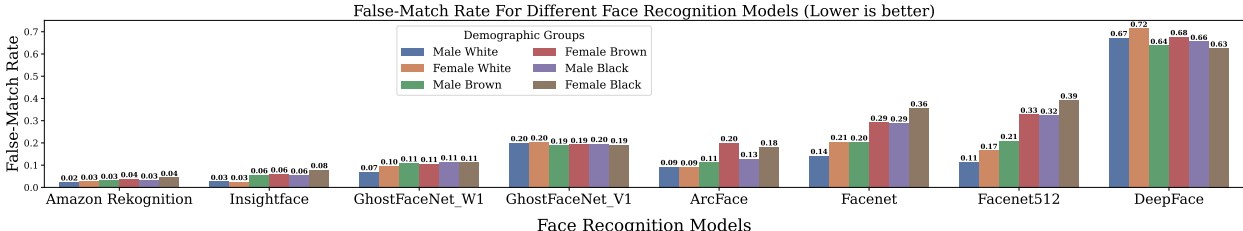

Figure 40: False-match rate (FMR) of all the face embedding models across the six demographic groups. Amazon Rekognition and InsightFace have the lowest FMR values. Moreover, these two models have lowest disparity of FMR over the demographic groups.

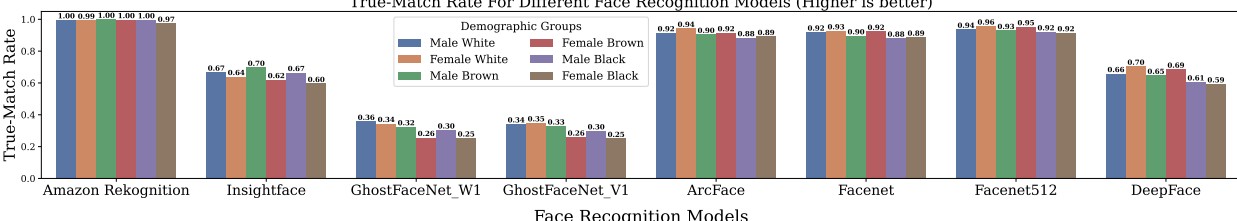

Figure 41: True-match rate (TMR) of all the face embedding models across the six demographic groups. Amazon Rekognition model has the highest TMR values.

Table 12: Distribution of caption counts for sampled entities in celebrities, politicians, and art styles domains.

| Caption Counts (LAION-2B) | Celebrities | Politicians | Classical Artists | Modern Artists |
|---|---|---|---|---|
| 0 | 19 | 15 | 14 | 15 |
| 1-100 | 48 | 60 | 67 | 69 |
| 100-500 | 57 | 120 | 133 | 139 |
| 500-1K | 52 | 80 | 62 | 62 |
| 1K-5K | 151 | 65 | 63 | 64 |
| 5K-10K | 19 | 40 | 39 | 32 |
| > 10K | 53 | 40 | 40 | 34 |

