# OpenReview forum: "How Many Images Does It Take? Estimating Imitation Thresholds in Text-to-Image Models"
_TMLR — Accepted by TMLR_

### Review · Reviewer_Jfgf · 2025-10-14

**Summary Of Contributions:**

This paper introduces the concept of an "imitation threshold" in text-to-image models. This threshold is defined as the number of training images of a particular concept (e.g., a person's face, an artist's style) required for a model to be able to imitate that concept in its generated images. The authors conduct experiments on two domains, human faces and art styles, using several text-to-image models and pre-training datasets. Their findings indicate that the imitation threshold is in the range of 200-700 images, depending on the specific domain and model. The authors argue that this imitation threshold can serve as an empirical basis for copyright and privacy violation claims and as a guideline for developers of text-to-image models.

**Audience:**

Yes

**Audience Explanation:**

The paper sits at the intersection of core machine learning research and urgent, real-world ethical and legal challenges, making it highly relevant for TMLR's topic.

**Broader Impact Concerns:**

The paper may inadvertently provide a "recipe" for imitation. By establishing a target threshold (200-700 images), the research gives malicious actors a clear, quantitative goal for harmful activities. Conversely, the findings could be misinterpreted by developers as a "safe harbor", leading to a reduction in diligence.

The authors should add a dedicated "Broader Impact" or "Ethical Considerations" section to their paper to address the concerns.

**Claims And Evidence:**

Yes

**Claims Explanation:**

The authors have successfully moved from plausible claims in the abstract to a well-supported argument in the main text by providing robust methodology, validation through human studies, and thoughtful analysis.

**Requested Changes:**

- Elaborate on the Generalizability of the Threshold.
The paper focuses on two important but specific domains: faces and art styles. In the Discussion/Limitations section, the authors could briefly elaborate on how they hypothesize the imitation threshold might change for other types of concepts. For instance, would the threshold for concrete objects (e.g., "a specific type of chair") be lower or higher than for abstract styles? What about complex scenes? A short, speculative discussion on this would enrich the paper's conclusion and highlight more avenues for future work.

- More Detailed Intuition about Proposing "Imitation".
The authors do a great job of operationalizing "imitation" using domain-specific embedders and validating it with human studies. To further strengthen the paper, I suggest adding a paragraph to the discussion that reflects on the inherent subjectivity of this concept. For example, the authors could discuss that while their method provides a robust and reproducible proxy, the legal and social definitions of imitation may have further nuances that are difficult to capture computationally. This would add a layer of valuable self-reflection to the work.

- Expand on Unexplained Outliers as Future Work.
The outlier analysis in Section 7 is a major strength. The authors successfully explain many outliers (e.g., aliases, multiple people per image). They also honestly state that they couldn't explain all of them. It would be beneficial to explicitly frame the investigation of these remaining outliers as a promising direction for future research in the Discussion section. What other factors (e.g., image quality, caption style, visual diversity within a concept) might explain why a concept with many training images still has a low imitation score?

---

> ### Author Response · Authors · 2025-10-31
> **Response to Review**
>
> We thank the reviewer for taking the time and effort to review our paper and **we are glad that the reviewer found our work to provide robust methodology for computing the imitation threshold as well as thoughtful analysis via human subject studies**. Overall we agree with all the three changes the reviewer has requested and comment on them below:
>
> > Elaborate on the Generalizability of the Threshold.
>
> The imitation thresholds we find in this work are domain specific, which we also discuss in the limitations of the work. We believe that specific entities (like a specific type of chair) would have a similar imitation threshold as a human face or art style, however complex scenes that require the model to learn interaction between entities would require a higher threshold. We see our methodology as one of the most important contributions of our paper, and as such, it could be applied to other domains of interest in future work.
>
> > I suggest adding a paragraph to the discussion that reflects on the inherent subjectivity of imitation.
>
> We are glad that the reviewer found our operationalization of imitation useful and well validated. And we agree with the reviewer that despite Mimetic being a robust and reproducible approach, there remain parts to the method that have inherent subjectivity, especially, its evaluation that uses human subject experiments. In these experiments we ask humans to judge if a generated image is an imitation of the real image of a concept and provide a rating between 1-5 for it. This combined with legal definitions of imitation add to the subjectivity of our approach and as suggested by the reviewer we will add them to the discussion section of the paper.
>
> > It would be beneficial to explicitly frame the investigation of these remaining outliers as a promising direction for future research in the Discussion section.
>
> We are also glad that the reviewer found our analysis of outliers valuable and honest. We definitely agree with the reviewer and we expand on these outliers as a promising avenue for future work in the discussion section.
>
> > the findings could be misinterpreted by developers as a "safe harbor" and may inadvertently provide a "recipe" for imitation
>
> We have mentioned in the discussion section of our work that the imitation threshold we find should not be directly interpreted as a guarantor of safety from imitation, we will further elaborate and emphasize this point, and discuss the risk of malicious imitation based on this information in the limitations section of the paper.

---

### Review · Reviewer_GVNC · 2025-10-21

**Summary Of Contributions:**

This paper investigates the "number of images of a concept that a text-to-image model requires for imitating it." They also propose the idea of predicting 'imitation thresholds' that, if accurate, could be potential empirical basis for copyright violation claims.

Strengths:
1. The idea of using imitation thresholds is interesting and has not yet been explored. It could prove useful for attribution and detecting the level of infringement.
2. They also validate their automatic imitation‐score metric via human subject studies (both for faces and art styles) to ensure correlation between their quantitative measure and human judgments.

Weaknesses:
1. Although the authors cover three versions of one line of text-to-image model (Stable Diffusion variants), they do not test across a broad array of models/architectures.
2. Although two domains (faces + art styles) are covered, the authors explicitly note that their results do not claim generalization to all domains/models/data. Hence, the imitation threshold is model- and domain-specific.
3. No explanation has been provided on how this methodology could be extended to other text-to-image models. It would be useful to discuss how this work could lead to a benchmark and a tool that would make it easier for evaluation.
4. How computationally demanding is this methodology? Please provide concrete compute estimates to clarify whether determining the imitation threshold is feasible within a reasonable timeframe for a given model. Is this analysis intended to be performed on a case-by-case basis, or could it be scaled to a broad, open-ended exploration across a large sample set—potentially requiring unbounded compute resources?

**Audience:**

Yes

**Audience Explanation:**

This work represents the first systematic attempt to quantitatively estimate the imitation threshold, yielding insights with potential implications for researchers in the field of Generative AI, large corporations building foundation models, AI policy makers, and ethics committees worldwide.

**Broader Impact Concerns:**

Given the well-documented issues of illegal, harmful, and biased content within the LAION datasets, the authors should address these limitations in the broader impact statement and outline how their approach could be applied to alternative models or datasets, instead of focusing exclusively on LAION.

Additionally, there could also be serious repercussions if the imitation score proves to be inaccurate.

**Claims And Evidence:**

Yes

**Claims Explanation:**

The authors apply their proposed method (MIMETIC²) to three different text-to-image models (specifically variants of Stable Diffusion: SD 1.1, SD 1.5, and SD 2.1) trained on two different pretraining datasets (LAION2B-en and LAION-5B).

**Requested Changes:**

Please check the weaknesses section above.

---

> ### Author Response · Authors · 2025-10-31
> **Response to Review**
>
> We thank the reviewer for taking the time and effort to review our work and **we are glad that the reviewer found our idea of using imitation thresholds novel and our decision to validate the imitation score with human subject studies sound**. We address the limitations below:
>
> > they do not test across a broad array of models/architectures.
>
> Our approach requires access to the training dataset and the image generation model. Stable Diffusion model variants are the only set of open-sourced models whose training data is also available and therefore we evaluated our approach on all fully open-source models that had their training data also available.
>
> > the authors explicitly note that their results do not claim generalization to all domains/models/data
>
> Since our approach is based on the ability of the model to generate images from a domain, the imitation thresholds we estimate are model and domain specific. However, our method is one of the most important contributions of our work, and it can be applied to any domain and any models.
>
> > No explanation has been provided on how this methodology could be extended to other text-to-image models
>
> This paper proposes an algorithm that estimates the imitation threshold. As such, one can use the same methodology on any text-to-image model. We demonstrated it on two domains and four models, but it can be extended to any other domain.
>
> > How computationally demanding is this methodology? Is this analysis intended to be performed on a case-by-case basis or could it be scaled to a broad, open-ended exploration across a large sample set—potentially requiring unbounded compute resources?
>
> Our approach requires generating images from a domain and measuring the similarity between real images and generated images. This operation is overall not that expensive. In Appendix P (page 35 of the paper) we provide an estimate of the compute we used for this work: We use 8 L40 GPUs to generate images for the all text-to-image models in our work. Overall, we use them for 16 hours per prompt, per dataset, per model to generate images. Our analysis can be scaled to a large set of concepts and is still cheap.

---

> ### Comment · Action_Editor_HP2V · 2025-11-24
> **Final recommendation**
>
> Hi Reviewer GVNC,
>
> We are behind in processing this manuscript. Could you please submit your final recommendation after reviewing the rebuttal, as soon as possible?
>
> Thanks AC

---

### Review · Reviewer_CMUh · 2025-10-26

**Summary Of Contributions:**

This paper studies how many training images are required for a text-to-image generative model to sufficiently imitate a given concept from its training data. This is an important question for understanding model behavior in the context of copyright and privacy, because it aims to quantify how much exposure is “enough” for a model to reproduce a concept.

## Strengthes
1. **Novel and well-motivated task.** The problem setup is, to my knowledge, new. It is directly relevant to auditing and regulating generative models, especially around privacy and copyright exposure. Studying how imitation emerges as a function of training frequency is both practically and societally meaningful.

2. **Methodological originality.** A naïve approach would require a costly greedy search over different training set sizes to locate the exact imitation threshold. Instead, the paper leverages the above assumptions to avoid brute-force search. The core idea is to sort concepts by frequency and identify a significant change point in imitation scores, interpreting that change point as an estimate of the required sample count. While the approach relies on several strong assumptions, the overall formulation is insightful and feels original.

3. **Thorough empirical analysis.** The paper provides extensive experiments and diagnostic analyses across multiple concepts, models, and datasets. This helps the reader assess robustness and generality. The outlier analysis is especially valuable because it surfaces where and why the method fails. The authors also attempt to validate each of their assumptions empirically, rather than leaving them unexamined.

## Weakness

**Weaknesses / Questions**

I do not see a single critical flaw that would invalidate the overall contribution. That said, several parts of the pipeline are doing important work (the assumptions, the discriminator, the definition of the “change point,” etc.), and some details are underspecified. Clarifying these would strengthen the paper:

1. **Experimental details of key components are missing.**
   - **Discriminator:** What is its performance (e.g., precision/recall, calibration)? Since the discriminator appears central to measuring “imitation,” its reliability matters.
   - **Change point detection:** How exactly is the significant change point defined and selected? Is it statistically motivated, heuristic, or tuned?

2. **Validation of assumptions could be deeper.**
   The paper does provide some supporting evidence for the three assumptions, but in several cases the argument feels high-level.
   - **Distributional invariance:** Instead of only reporting an average difference, it would help to show the full distribution of imitation scores (or score deltas) across concepts. This would let us judge variance and tail behavior.
   - **No confounders:** The text states, “we assume there are no confounders between imitation score and the image count of a concept, i.e., the imitation score … is not affected by the presence of other concepts in the training data.” This equivalence is not entirely clear. The “image count of a concept” and “presence of other concepts” are not obviously the same variable. Please clarify the causal argument here. For example: how does the fine-tuning experiment isolate the effect of image count alone?
   - **Equal contribution per image:** Similarly, showing distributions (not just averages) of per-image contribution would make this claim more convincing.

3. **Ambiguity in Table 7.**
   Table 7 is hard to interpret in its current form. It is unclear what the reported numbers represent (e.g., thresholds? error? counts? correlation measures?). A clearer caption and/or column descriptions would help.

4. **Imitation threshold depends on generative models and datasets**. From table 3, we can see the results vary with different generative models and datasets. Therefore, it's still hard to use it as a universal standard for all image generators.

**Additional Comments:**

This reviewer is not an expert in image generation.

**Audience:**

Yes

**Audience Explanation:**

I believe this propose method and the findings can benefit the community and even broader audience. The analysis help understanding the behaviors of generative models, and establishing regulerization for privacy and copyright protection.

**Broader Impact Concerns:**

Since this work is to analyze the generative model behaviors, not generating content, I did not see big concerns.

**Claims And Evidence:**

Yes

**Claims Explanation:**

In general, I think the claims made in the submission are well-supported by empirical experiments. The authors conducted experiments to validate their assumptions, analyze outliers, as well as experiments across different models, concepts, and artists.

**Requested Changes:**

- **Provide quantitative details for core components (CRITICAL).**
  - Report the discriminator’s performance (e.g., precision/recall, calibration, typical failure modes), since this model underlies how “imitation” is measured.
  - Clearly define how the “significant change point” in imitation score is detected (formal criterion, threshold, or statistical test).
  - These details are necessary to assess reproducibility and to judge whether the estimated sample-count threshold is trustworthy.

- **Clarify the causal claim behind the “no confounders” assumption (CRITICAL).**
  - The paper assumes that a concept’s imitation score depends only on that concept’s own image count and is not affected by the presence of other concepts.
  - Please clarify:
    1. How the fine-tuning experiments isolate the effect of image count alone.
    2. In what sense “image count of a concept” and “presence of other concepts” are being treated as equivalent factors, as stated in the last second paragraph in Section 3.
  - This is critical because this assumption is directly used to argue that the estimated threshold is meaningful.

- **Improve presentation of assumption validation (IMPORTANT / borderline critical).**
  - For *distributional invariance*: instead of only reporting an average difference, show the full distribution of imitation scores (or score deltas) across concepts, so we can see variance and tails.
  - For *equal per-image contribution*: similarly, show distributions (not just means) of per-image contribution.
  - These additions would significantly strengthen confidence in the assumptions, and therefore in the method.

- **Clarify Table 7 (IMPORTANT).**
  - Table 7 is currently difficult to interpret. Please explain in the caption (or main text) what each number represents and how to read the table.
  - While not strictly critical for acceptance on its own, this clarification is needed to fully understand part of the empirical results.

---

> ### Author Response · Authors · 2025-10-31
> **Response to Review**
>
> We thank the reviewer for taking the time and effort to provide a detailed review of our work. We are glad that the reviewer found our task to be novel and well-motivated, and method original, and our analyses thorough. We address the limitations below:
>
> > Report the discriminator’s performance (e.g., precision/recall, calibration, typical failure modes)
>
> Please see the table below. Notice that we have reported all these numbers in the appendix, and we include the exact reference to their location in this table. If you think it is important, we can include this table in the paper.
>
> | Domain           | Accuracy (%) | Precision (%) | Recall (%) | In paper             |
> |------------------|--------------|---------------|-------------|----------------------|
> | Celebrity        | 100          | 100           | 100         | Figure 7 page 22     |
> | Politicians      | 100          | 100           | 100         | Figure 7 page 22     |
> | Classical Artists| 95.0           | 92.3          | 98.3        | Figure 12 (a) page 25|
> | Modern Artists   | 94.4         | 92.9          | 96.1        | Figure 12 (b) page 25|
>
> > Clearly define how the “significant change point” in imitation score is detected (formal criterion, threshold, or statistical test)
>
> The detection of change point is a statistical test based using the PELT algorithm [Killick et al., 2012]. PELT is a general algorithm that detects a statistical change in a series of points.
> It does this by splitting the data into segments while keeping only the most useful change points. We describe this in detail in Section 4.3 in the paper. This paper describes the algorithm: https://arxiv.org/abs/1101.1438
>
> > “no confounders” assumption: How the fine-tuning experiments isolate the effect of image count alone.
>
> Since we want to isolate the influence of any related concept on a particular concept’s imitation threshold, we find if the imitation threshold found during pre-training (the main results in the paper) and the imitation threshold found during finetuning are the same. The finetuning experiment isolates the effect of image count because of the experimental design in which (a) we only consider concepts that the model has not seen during pre-training and thus the unique knowledge of that concept has to come from the finetuning experiment only, and (b) we finetune on images of that new entity and if any other concept has influence on the imitation thresholds it must be the same in pre-training and finetuning case as we find the imitation thresholds to be the same in both cases .
>
> >“no confounders” assumption: In what sense “image count of a concept” and “presence of other concepts” are being treated as equivalent factors
>
> “Image count of a concept” and “presence of other concepts” are not equivalent factors and we do not make this assumption. Our assumption states that: “there are no confounders between the imitation score and the image count of a concept, i.e, the imitation score of a concept is not affected by the presence of other concepts in the training data”. We want to isolate the influence of image count of a concept on its imitation threshold for which we conduct the finetuning experiment (as explained above).
>
> > For distributional invariance: instead of only reporting an average difference, show the full distribution of imitation scores
>
> Here is the distribution of differences in imitation scores: https://imgur.com/a/rVqpevO
>
> > For equal per-image contribution: similarly, show distributions (not just means) of per-image contribution.
>
> Here is the distribution of imitation scores for this assumption: https://imgur.com/a/6hXS7XB
>
> > Table 7: Please explain what each number represents and how to read the table.
>
> The 4 columns in Table 7 present statistics of images we missed due to the assumption that a person appears in an image only if they are mentioned in the caption. Therefore, we quantified how often a person is found in the image but not mentioned in the caption. Since this is most likely to happen for popular individuals, we search for such individuals (the first column in Table 7) and report the number of captions they are mentioned in. As our results suggest, our assumption is not far off from the truth, and we only miss a very small percentage of the images, compared to our estimates (third column in the table). We also calculate the estimates of the total missed images based on the 100K sample size (the last column). We will add a summary to the caption of Table 7, and an elaborated discussion to the corresponding section.

---

> ### Comment · Action_Editor_HP2V · 2025-11-24
> **Final recommendation**
>
> Hi Reviewer CMUh,
>
> We are behind in processing this manuscript. Could you please submit your final recommendation after reviewing the rebuttal, as soon as possible?
>
> Thanks AC

---

> > ### Comment · Reviewer_GVNC · 2025-11-24
> > **Apologies for the delay**
> >
> > Hi AC,
> >
> > Sorry for the delay. I have submitted my recommendation.

---

### Decision · Action_Editor_HP2V · 2025-11-29

**Recommendation:** Accept with minor revision

**Audience:**

Yes

**Audience Explanation:**

Due to the novelty of the topic and the scarcity of existing work, the reviewers and AE agree that this study should be of interest to the ML sub-community focused on generative modeling such as text-to-image and text-to-video diffusion models.

**Claims And Evidence:**

Yes

**Claims Explanation:**

This submission makes two main claims:
1. a new problem setup for estimating how many images are required to learn a concept in text-to-image models; and
1. a method, MIMETIC², together with its empirical validation on face and art-style data using LDM and Stable Diffusion, to estimate such a learning threshold.

According to the reviewers, the first claim is well-substantiated and is viewed as a strength of the paper. They acknowledge the value of the proposed problem formulation for this timely topic. Regarding the second claim, which is about the proposed method and its validation, received more scrutiny, with concerns about the limited domain tested and questions regarding its generalization beyond latent diffusion models, such as velocity-prediction variants, or the generative flow models.

The AE believes that the main claims are reasonably supported, especially considering this work as an initial step in a new and inherently complex research direction. Despite the noted limitations, the authors have made meaningful efforts to address the challenges to design their method. The work provides a baseline of sound quality and reasonable empirical analysis for future research to build upon.


However, the authors need to address their claim that the methodology can be readily extended to other T2I image models. A thorough discussion, accompanied by empirical evidence supporting this claim, is expected.